# GTA: A Benchmark for General Tool Agents

**Jize Wang**[1,2] **Zerun Ma**[2] **Yining Li**[2] **Songyang Zhang**[2]
**Cailian Chen**[1] **Kai Chen**[2*] **Xinyi Le**[1*]

[1]Shanghai Jiao Tong University [2]Shanghai AI Laboratory
{jizewang2000,cailianchen,lexinyi}@sjtu.edu.cn
{mazerun,liyining,zhangsongyang,chenkai}@pjlab.org.cn

## Abstract

Significant focus has been placed on integrating large language models (LLMs) with various tools in developing general-purpose agents. This poses a challenge to LLMs' tool-use capabilities. However, there are evident gaps between existing tool-use evaluations and real-world scenarios. Current evaluations often use AI-generated queries, single-step tasks, dummy tools, and text-only interactions, failing to effectively reveal the agents' real-world problem-solving abilities. To address this, we propose GTA, a benchmark for **G**eneral **T**ool **A**gents, featuring three main aspects: (i) *Real user queries*: human-written queries with simple real-world objectives but implicit tool-use, requiring the LLM to reason the suitable tools and plan the solution steps. (ii) *Real deployed tools*: an evaluation platform equipped with tools across perception, operation, logic, and creativity categories to evaluate the agents' actual task execution performance. (iii) *Real multimodal inputs*: authentic image files, such as spatial scenes, web page screenshots, tables, code snippets, and printed/handwritten materials, used as the query contexts to align with real-world scenarios closely. We design 229 real-world tasks and executable tool chains to evaluate mainstream LLMs. Our findings show that real-world user queries are challenging for existing LLMs, with GPT-4 completing less than 50% of the tasks and most LLMs achieving below 25%. This evaluation reveals the bottlenecks in the tool-use capabilities of current LLMs in real-world scenarios, which provides future direction for advancing general-purpose tool agents. Dataset and code are available at https://github.com/open-compass/GTA.

## 1 Introduction

Integrating tools with large language models (LLMs) has attracted broad research interest as a potential approach towards general AI assistants. Notable works include LangChain [5], AutoGPT [8], and ChatGPT Plugins [19]. These systems decompose workflow into two interactive parts: planning and execution, respectively handled by LLM controllers and callable tools. Solving complex real-world tasks requires multiple types of tools, including perception, operation, logic, and creativity, posing great challenges to LLMs' tool-use proficiency. Consequently, evaluating the models' tool-use capabilities for real-world tasks is crucial for enhancing the effectiveness of agent systems.

Despite the progress on benchmarking the tool-use capability of LLMs made by recent works, especially on collecting massive APIs and AI-generated user queries to enable scalable testing, there remain noticeable gaps regarding real-world scenarios, as shown in Table 1. First, AI-generated user queries, limited by the generative model, often result in overly brief or monotonous solutions.

---

*Corresponding Authors.

38th Conference on Neural Information Processing Systems (NeurIPS 2024) Track on Datasets and Benchmarks.

Table 1: Comparison of benchmarks for the LLM-based agent system. *Real-world means solving the queries is helpful for humans in real life while step-implicit and tool-implicit for LLMs.

| Method | Real-world* user quries | Real deployed tools | Multimodal context inputs | Human annotated tool chains | Execution result evaluation |
|---|---|---|---|---|---|
| APIBench [21] | | | | | |
| ToolBench [23] | | ✓ | | | |
| APIBank [12] | | ✓ | | ✓ | |
| GAIA [16] | ✓ | | ✓ | | ✓ |
| m&m's [14] | | ✓ | ✓ | ✓ | ✓ |
| **GTA (Ours)** | ✓ | ✓ | ✓ | ✓ | ✓ |

Figure 1: Some samples in the GTA benchmark. The user queries are human-designed, step-implicit, tool-implicit, and settled in real-world scenarios. Multimodal context inputs are provided. Solving these queries is helpful for users and complex for a LLM-based tool agent. The agent must use a combination of executable tools in perception, operation, logic, and creativity categories.

This is unsuitable for evaluating agent systems' reasoning and planning capability, as shown in Table 2. Second, existing tool-use benchmarks mainly focus on text-formed user-agent interaction, lacking assessment of multimodal capabilities, thus falling short of aligning with real-world scenarios effectively. Third, existing tool-use evaluation approaches build up virtual tools. They can only evaluate isolated steps in the tool invocation chains, thus unable to reflect the agents' capability to accomplish complex tasks end-to-end.

To ensure the evaluation closely reflects real-world scenarios, we consider the authenticity of user queries, tools, and interaction modalities. We propose a comprehensive tool-use evaluation with real-world user queries. The primary features of the evaluation are:

i. ***Real user queries.*** The user queries are designed by humans, rather than generated by AI, to reflect real-world tasks accurately. These queries describe tasks with clear objectives, but the tool-use steps are implicit. Thus, the LLM must use reasoning to deduce the suitable tools to address the given tasks. In this way, we avoid the drawbacks of using AI-generated queries in which the tool invocation steps are often explicitly hinted at. Moreover, each query requires multiple steps to resolve, necessitating the model to plan the sequence of tool invocations.

ii. ***Real deployed tools.*** We provide an evaluation platform deployed with tools across various categories, such as perception, operation, logic, and creativity. All tools are executable rather than simulated by text description. A detailed and executable ground truth tool chain is provided for each task, including each tool-use step and the final answer. Each step includes the tool name,

Table 2: Comparison of GTA queries with AI-generated queries. The steps and tool types for queries in ToolBench and m&m's are explicitly stated, as marked in red and blue. The queries in APIBench are simple and only contain one step. Our GTA's queries are both step-implicit and tool-implicit.

| Method | Queries |
|---|---|
| ToolBench | Need to create an ASCII art representation of a mathematical equation. The equation is... Help me generate the ASCII art... **Also** please **generate an ASCII art representation** of the text... (**Related tools**: figlet, list figlet styles, matheq) |
| APIBench | Our customer is a zoo and we want to help them **detect movement** of different animals. Write a Python program in 1 to 2 lines to call API in TensorFlowHub. (**Related tools**: ObjectDetection) |
| m&m's | I need an illustration for my children's book. I've imagined a scene where there's a large group of little kids... **After** we have the image, we also need to **identify all the objects**, **then add labels** to them. (**Related tools**: ImageGeneration, ObjectDetection, Tagging) |
| **GTA (Ours)** | Convert the table into a statistical chart with the type of image shown in the example. (**Related tools**: ImageDescription, OCR, Plot) |

     argument value, and return value. The detailed tool chains enable a fine-grained evaluation of the actual problem-solving abilities of tool agents.

iii. ***Real multimodal inputs.*** Each query is accompanied by one or two authentic image files, including spatial scenes, webpage screenshots, tables, code snippets, printed/handwritten materials, etc., to serve as the context for the user queries. The LLM is required to solve the problem based on the multimodal context and user queries. This setting closely aligns with the multimodal real-world problem-solving scenarios.

We manually design 229 real-world tasks and corresponding executable tool chains to evaluate mainstream LLMs. We build a platform covering a total of 14 tools across perception, operation, logic, and creation categories. Tools and some data samples are illustrated in Figure 1. We design fine-grained tool evaluation metrics that cover the entire process of tool invocation. Our findings indicate that real-world scenario queries present challenges to existing LLMs, with GPT-4 completing fewer than 50% of the tasks and most LLMs managing less than 25%.

In summary, our contributions are as follows:

- A tool-use benchmark for general tool agents. The user queries are human-designed, step-implicit, and settled in real-world scenarios. Multimodal contextual inputs are provided. Each query has a corresponding executable tool chain to enable a fine-grained tool-use evaluation.
- An evaluation platform equipped with a wide variety of executable tools covering the categories of perception, operation, logic, and creativity. Fine-grained metrics are designed for tool use, unveiling tool-augmented LLMs' reasoning and planning capabilities in real-world scenarios.
- Evaluation and analysis of mainstream large language models. We evaluate the tool-use ability of 16 LLMs in multiple dimensions. Our findings reflect the tool-use bottleneck of existing LLMs in real-world scenarios, providing suggestions for the development path of general tool agents.

## 2 Related Work

**LLM-based agents.** In the pursuit of developing general-purpose agents, there has been considerable focus on integrating LLMs with external tools. These LLM-based agents enable powerful capabilities in environment interaction, decision-making, and task execution. Open-source platforms have been proposed, such as LangChain [5], AutoGPT [8], and BabyAGI [17]. Moreover, several efforts have been made to achieve specialized capabilities by integrating specialized tools into LLMs. WebGPT [18], WebCPM [22], WebShop [32] are proposed to enhance the model's web search ability. RestGPT [26] combines LLM with RESTful APIs to enable web service development. In the visual domain, Visual ChatGPT [29], MM-ReAct [31], MLLMtool [28], and LLaVA-Plus [13] prompt or finetune LLMs to interact with visual models. In the data analysis domain, DataCopilot [36] manages and processes massive data autonomously by invoking data analysis tools. HuggingGPT [24], ModelScopeAgent [11] build agent systems using LLMs integrated with massive machine learning models. In the field of human-computer interaction, AppAgent [35] allows LLMs to mimic human stapping and swiping operations to operate smartphones. In these works, the LLM serves as a central

controller, invoking a certain class of tools to accomplish specialized tasks. In real-world scenarios, the environment is more complex. This requires LLMs to engage in planning and coordination among various types of tools, thereby posing a challenge to their tool-use capabilities.

**Tool-use evaluations.** With the rise of LLM-based agents, many studies have been conducted to evaluate the tool-use capabilities of LLMs. ToolBench [23] collects RESTful APIs and leverages ChatGPT [1] to design tool-use tasks and corresponding tool chains. Two metrics, Pass Rate and Win Rate, are devised to evaluate the efficacy of tool use. APIBench [21] is a comprehensive dataset that includes APIs from HuggingFace, TorchHub, and TensorHub, with evaluation metrics focusing on Abstract Syntax Tree (AST) accuracy. API-Bank [12] comprises 53 commonly utilized APIs, such as SearchEngine, PlayMusic, BookHotel, and ImageCaption, along with a comprehensive tool-augmented LLM workflow to evaluate the API calling, retrieving, and planning abilities. m&m's [14] is a benchmark to evaluate tool use for multi-step multimodal tasks. It aims to evaluate different planning strategies for LLMs as planning agents. Most of the benchmarks above, however, rely on AI-generated queries. The tool-use steps are explicitly and rigidly included. Thus, these queries do not accurately represent real-world scenarios. Among many previous studies, GAIA [16] is renowned for its real-world scenario based benchmark aiming at evaluating general AI assistants, which is closer to our work. It designs conceptually simple questions for humans yet is challenging for most advanced AIs. However, GAIA focuses on artificial general intelligence (AGI). In contrast, GTA is designed to evaluate tool agents specifically, offering real-deployed tools and executable tool chains for a fine-grained evaluation in real-world scenarios. Osworld [30] is also a real-world benchmark featuring multi-step, complex tasks inspired by authentic user cases. Still, it is specifically tailored for computer environments, whereas GTA is devised for tool agents operating in more generalized real-world scenarios.

## 3 GTA Benchmark

In this section, we describe the design and content of GTA. The whole dataset construction pipeline is shown in Figure 2. We first present the composition of each sample in the dataset in Section 3.1. The construction method of queries and tool chains are depicted in Section 3.2 and Section 3.3, respectively. We then present the dataset's statistics in Section 3.4.

### 3.1 Dataset Formulation

Given a set of tools $\mathcal{T}_c = \{t_k\}_{k=1}^N$, a sample in GTA is composed of five parts $(\mathcal{F}, \mathcal{Q}, \mathcal{T}, \mathcal{C}, \mathcal{A})$. Among these parts, $\mathcal{F}$ is a set of files containing one or two images. $\mathcal{Q}$ is a query based on $\mathcal{F}$. It is a real-world scenario based problem of simple form but needs to be solved through multiple steps with tools in $\mathcal{T}_c$. Which tools need to be used, and in what steps are not explicitly included in the query. They require reasoning and planning by the LLM, which serves as a central controller. This procedure is given in the reference tool chain $\mathcal{C} = \{s_i\}_{i=1}^m$. The tool chain contains $m$ steps. Each step is $s_i = (t_i, a_i, r_i)$, where $t_i$ is the tool used in step $i$. $a_i$ and $r_i$ indicate arguments and return values. $\mathcal{T} = \bigcup_{j=1}^m \{t_j\} \subseteq \mathcal{T}_c$ notes the set of tools involved in this query. $\mathcal{A}$ is the final answer yielded by the LLM after reasoning with tools.

In our setting, $\mathcal{T}_c$ contains 14 tools across four categories, including perception, operation, logic, and creativity. The full list of tools is shown in Figure 1, and more detailed information can be found in Appendix B.1. The queries $\mathcal{Q}$ are classified into three types: subjective, objective, and image generation. Examples of the three types of queries are shown in Appendix B.2. For a subjective query $\mathcal{Q}_s$, the final answer $\mathcal{A}$ is usually some descriptive text. It is not unique, but the general idea is the same. In this case, $\mathcal{A}$ contains a list of three reference answers. For an objective query $\mathcal{Q}_o$, $\mathcal{A}$ is a uniquely determined number or phrase. For an image generation query $\mathcal{Q}_g$, we do not measure the generated image directly. In this situation, $\mathcal{A} = \emptyset$.

### 3.2 Query Construction

To construct $(\mathcal{F}, \mathcal{Q}, \mathcal{T})$, we first gather human-designed queries that meet three main principles: **i)** Given $\mathcal{T} \subseteq \mathcal{T}_c$, the task $(\mathcal{F}, \mathcal{Q})$ can be solved with the capabilities enabled by tools in $\mathcal{T}$. **ii)** To evaluate LLMs' reasoning and planning abilities, the tool invocation steps should not be explicitly

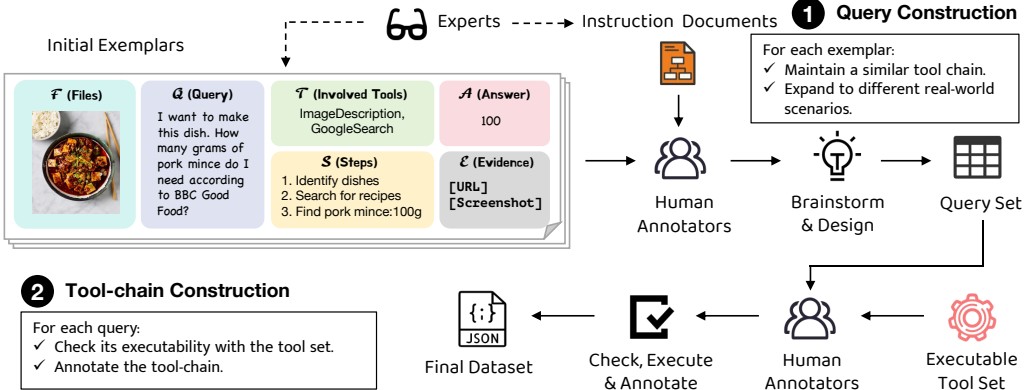

Figure 2: Two steps are performed in the dataset construction pipeline. ❶ During *query construction*, initial exemplars and instruction documents are designed by experts and given to human annotators. Annotators brainstorm and design more samples based on the exemplars. ❷ During *tool chain construction*, annotators manually call the deployed tools to check the executability of each query in the query set. Then they annotate the ground truth tool chains for each query.

stated in the queries. **iii)** The queries are meaningful and based on real-world scenarios. Satisfying all the principles simultaneously is challenging. It requires $\mathcal{F}$, $\mathcal{Q}$, and $\mathcal{T}$ to match each other sensibly and logically. We use a query construction pipeline based on exemplar expansion, as shown in the first part of Figure 2. We first give some initial exemplars with diverse scenarios and tool combinations. Then, we instruct annotators to create more queries based on the exemplars.

**Exemplar designed by experts.** We first design some initial questions as exemplars, which are provided in Appendix C.1. These example questions are of diverse scenarios and contain different tool combinations. Every sample should comprise six components: $\mathcal{F}$ (image files), $\mathcal{Q}$ (queries), $\mathcal{T}$ (involved tools), $\mathcal{S}$ (solution steps), $\mathcal{A}$ (answers), and $\mathcal{E}$ (evidence). Image files $\mathcal{F}$ could be obtained from the internet, and their URLs must be recorded. $\mathcal{F}$ could also be a photo taken or a diagram drawn by the annotators. The query $\mathcal{Q}$ must avoid obvious references to a specific tool. For example, the query *please describe the image for me* is unqualified since it obviously refers to the tool ImageDescription. The components $\mathcal{S}$, $\mathcal{A}$, and $\mathcal{E}$ will not appear in the final dataset but are utilized to assist annotators in meeting the annotation requirements. $\mathcal{S}$ represents the steps required to solve the problem. Annotators should note down the steps, ensuring their number exceeds two. The answer $\mathcal{A}$ of objective queries should be given to guarantee a unique answer. To ensure the uniqueness, the answer should not depend on the images generated in previous steps. For example, the question *what kind of animal is in the picture* should not be asked after *generate an image of an animal*, as the answer is uncertain. For queries utilizing the Google Search tool, $\mathcal{E}$ should include the answer's URL and a screenshot pinpointing the answer's location to verify the query's searchability with the tool.

**Diversified expansion by annotators.** After the initial exemplars are given, we instruct annotators to create more samples based on each exemplar. We adopt a diversified expansion strategy for the annotators to expand the questions based on the exemplars. The general idea is to keep the tool set $\mathcal{T}$ of the template unchanged or slightly modify it. Then, annotators brainstorm scenarios different from the template. Further information on the diversified expansion approach is detailed in Appendix C.2. For each sample, we have crafted a manual expansion example to serve as guidance for the annotators. After the expansion process, we perform a quality check and manually filter out the questions that do not satisfy the expansion requirements. The instruction documents for annotators are reported in Appendix C.3.

**Considerations for the search tool.** Web search tools like Google Search may return variable results over time, harming the accuracy and stability of evaluation. To address this problem, we perform two constraints. First, the question must be answered using web search rather than relying on an LLM's internal knowledge. This can be achieved by designing time-sensitive questions, such as *what is the 2023 QS ranking of Tsinghua University* rather than general inquiries like *where is Tsinghua University located in China*. We may also direct the query to a specific information source, for instance, asking *what is the recipe for Ma Po Tofu according to the BBC Good Food website* instead

| Item | Number |
|------|--------|
| Total query | 229 |
| Query w/ pure text answers | 172 |
| Query w/ image answers | 57 |
| Total tool calls | 557 |
| Image files | 252 |
| Tools | 14 |
| 1/2/3/4-tool examples | 17/147/50/15 |

Table 3: Basic statistics of GTA.

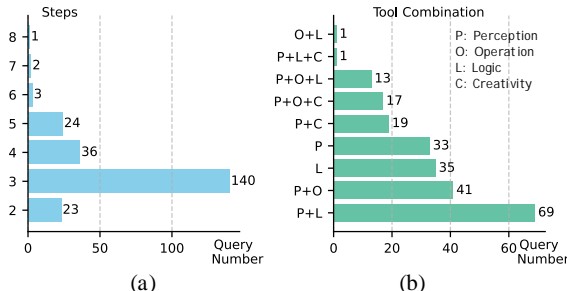

Figure 3: Other statistics of GTA. (a) Step number per query. (b) Frequency of different tool combination.

of a broad question like *what is the recipe for Ma Po Tofu*. Second, it is crucial to ensure that answers remain constant over time within an evaluation dataset. To fulfill this criterion, we can specify a time frame, web page, or organization within the question. An example would be *what is the 2024 QS ranking of Tsinghua University*, rather than *what is the QS ranking of Tsinghua University*.

### 3.3 Tool Chain Construction

Based on the $(\mathcal{F}, \mathcal{Q}, \mathcal{T})$ samples constructed in Section 3.2, we instruct three annotators majoring in computer science to manually construct the corresponding tool chain $\mathcal{C}$ and the final answer $\mathcal{A}$. We design a JSON file structure containing the query-related tool list, image paths, and ReAct [33] style dialog sequences. The dialog sequences include the user query, the executable tool chain, and the final answer. Initially, $(\mathcal{T}, \mathcal{F}, \mathcal{Q})$ are put into the associated sections for tools, images, and user queries. Subsequently, we deploy all tools in $\mathcal{T}_c$. The annotators utilize the tools according to the reference steps $\mathcal{S}$ and get the outcomes. They record this process in the tool chain section of the dialog sequences, alongside the final answer. Since we do not evaluate the tools' efficacy, when a tool fails to provide accurate recognition for a query (for instance, OCR inaccuracies in text recognition within diagrams), we discard the query. Through the above process, we ensure the feasibility of the questions, the executability of the tool chains, as well as the precision of the final answers. The structure of the tool chain is provided in Appendix C.4.

### 3.4 Dataset Statistics

GTA comprises a total of 229 questions, with the basic dataset statistics presented in Table 3. The dataset involves 252 images and 14 distinct tools. It includes 156 objective, 16 subjective, and 57 image generation queries. The number of tools involved in each question varies from 1 to 4, with most questions using 2 or 3 tools. The steps to resolve the questions range from 2 to 8, with most questions requiring 2 to 4 steps, as depicted in Figure 3(a). The detailed frequency distribution of different tool combinations is listed in Figure 3(b). P, O, L, C are short for Perception, Operation, Logic, Creativity, respectively. Perception+Logic and Perception+Operation are the most frequently appearing tool combination types.

## 4 Evaluation and Analysis

### 4.1 Experiment Settings

We evaluate 16 LLMs on GTA. For API-based models, we select GPT-3.5 [20], GPT-4 [1], GPT-4o, Claude-3 [2], and Mistral-Large [9]. For open-source models, we select Llama-3 [15] series, Qwen1.5 [3] series, Mistral [9], Mixtral [10], Yi [34] series, Deepseek [4] series. Experiments are conducted using NVIDIA A100 GPU within OpenCompass [7] evaluation platform. We adopt Lagent [27] as the agent framework. ReAct [33] is used as the tool invocation prompt schema. More experiment information can be found in Appendix D.1 and D.2.

We evaluate the models in two modes. **Step-by-step mode** is designed to evaluate the model's fine-grained tool-use capabilities. In this mode, the model is provided with the initial $n$ steps of the reference tool chain as prompts, with the expectation to predict the action in step $n + 1$. This method does not involve the actual use of the tool, and the prediction of each step does not depend on the

Table 4: **Main results of GTA.** Inst., Tool., Arg., Summ., Ans., Ans.+I denote InstAcc, ToolAcc, ArgAcc SummAcc, AnsAcc, and AnsAcc w/ ImgGen respectively. P., O., L., C. denote the F1 score of tool selection in Perception, Operation, Logic, and Creativity categories. **Bold** denotes the best score among all models. Underline denotes the best score under the same model scale. **AnsAcc** reflects the overall performance.

| Model | STEP-BY-STEP MODE | | | | END-TO-END MODE | | | | | |
|---|---|---|---|---|---|---|---|---|---|---|
| | Inst. | Tool. | Arg. | Summ. | P. | O. | L. | C. | Ans. | Ans.+I |
| *API-based* | | | | | | | | | | |
| GPT-4-1106-Preview | 85.19 | 61.4 | **37.88** | 75 | 67.61 | 64.61 | 74.73 | 89.55 | **46.59** | **44.9** |
| GPT-4o | **86.42** | **70.38** | 35.19 | 72.77 | **75.56** | **80** | **78.75** | 82.35 | 41.52 | 40.05 |
| GPT-3.5-Turbo | 67.63 | 42.91 | 20.83 | 60.24 | 58.99 | 62.5 | 59.85 | **97.3** | 23.62 | 21.18 |
| Claude-3-Opus | 64.75 | 54.4 | 17.59 | **73.81** | 41.69 | 63.23 | 46.41 | 42.1 | 23.44 | 14.47 |
| Mistral-Large | 58.98 | 38.42 | 11.13 | 68.03 | 19.17 | 30.05 | 26.85 | 38.89 | 17.06 | 11.94 |
| *Open-source* | | | | | | | | | | |
| Qwen1.5-72B-Chat | 48.83 | 24.96 | 7.9 | 68.7 | 12.41 | 11.76 | 21.16 | 5.13 | 13.32 | 10.22 |
| Mixtral-8x7B-Instruct | 28.67 | 12.03 | 0.36 | 54.21 | 2.19 | 34.69 | 37.68 | 42.55 | 9.77 | 9.33 |
| Deepseek-LLM-67B-Chat | 9.05 | 23.34 | 0.18 | 11.51 | 14.72 | 23.19 | 22.22 | 27.42 | 9.51 | 7.93 |
| Llama-3-70B-Instruct | 47.6 | 36.8 | 4.31 | 69.06 | 32.37 | 22.37 | 36.48 | 31.86 | 8.32 | 6.25 |
| Yi-34B-Chat | 23.73 | 10.77 | 0 | 34.99 | 11.6 | 11.76 | 12.97 | 5.13 | 3.21 | 2.41 |
| Qwen1.5-14B-Chat | 42.25 | 18.85 | 6.28 | 60.06 | 19.93 | 23.4 | 39.83 | 25.45 | 12.42 | 9.33 |
| Qwen1.5-7B-Chat | 29.77 | 7.36 | 0.18 | 49.38 | 0 | 13.95 | 16.22 | 36 | 10.56 | 7.93 |
| Mistral-7B-Instruct | 26.75 | 10.05 | 0 | 51.06 | 13.75 | 33.66 | 35.58 | 31.11 | 7.37 | 5.54 |
| Deepseek-LLM-7B-Chat | 10.56 | 16.16 | 0.18 | 18.27 | 20.81 | 15.22 | 31.3 | 37.29 | 4 | 3.01 |
| Llama-3-8B-Instruct | 45.95 | 11.31 | 0 | 36.88 | 19.07 | 23.23 | 29.83 | 42.86 | 3.1 | 2.74 |
| Yi-6B-Chat | 21.26 | 14.72 | 0 | 32.54 | 1.47 | 0 | 1.18 | 0 | 0.58 | 0.44 |

model's preceding outputs. This enables an alignment comparison between the model's output with each step of the ground truth tool chain. **End-to-end mode** is designed to reflect the tool agent's actual task executing performance dynamically. In this mode, the model actually calls the tools and solves the problem by itself. Each step relies on the preceding step's output. We compare the tools selected and the execution result with the ground-truth tool set and result under this mode.

## 4.2 Evaluation Metrics

We design fine-grained metrics spanning from the LLM's tool invocation process to execution results. To evaluate the tool invocation process, we devise four metrics under step-by-step mode: ***InstAcc***, ***ToolAcc***, ***ArgAcc***, and ***SummAcc***. InstAcc is instruction-following accuracy, which quantifies the percentage of steps executed without errors. ToolAcc measures the accuracy of tool selection. ArgAcc accesses the accuracy of argument name prediction. SummAcc reflects how accurately the model can summarize the final answers considering all previous tool-use steps. For end-to-end mode, we use ***AnsAcc*** to measure the accuracy of the execution result. Besides, we calculate the ***F1 scores of tool selection*** in perception, operation, logic, and creativity categories. The four F1 scores compare the model's tool selection with the ground truth tool set, measuring its tool selection ability.

In calculating the metric AnsAcc, we exclude image generation queries and focus solely on queries with pure text answers, including subjective and objective queries. For objective queries, the ground truth contains both a whitelist and a blacklist of phrases. An answer is considered correct if it includes all terms from the whitelist and excludes all terms from the blacklist. In the case of subjective queries, the ground truth contains three manually labeled responses from distinct annotators. We compute the cosine similarity (ranging from 0 to 1) between the model's prediction and each of the three ground truth answers, ultimately considering the highest score obtained. We also design a metric ***AnsAcc w/ ImgGen***, to take image generation queries into account indirectly. Given that the outcome of the image generation is determined solely by the input parameters, we evaluate the accuracy of these parameter predictions. If the predicted parameters are correct, the images produced should align with the specified task objectives. The specific score calculation formulas of subjective and image generation queries are shown in Appendix D.3.

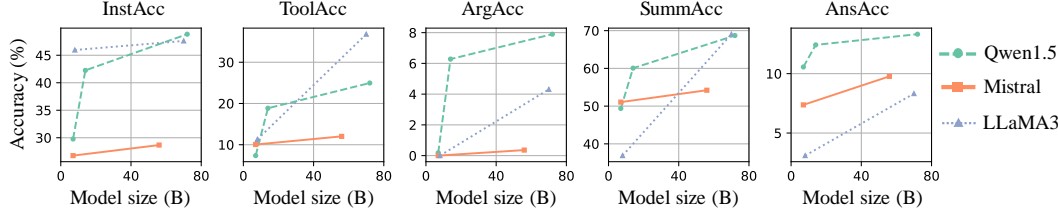

Figure 4: Performance of models with various size. Larger models within the same series perform better than their smaller counterparts, but larger models from different series do not necessarily outperform the smaller ones.

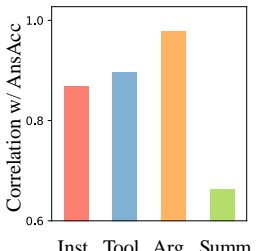

Figure 5: The Pearson correlation coefficient between AnsAcc and four metrics.

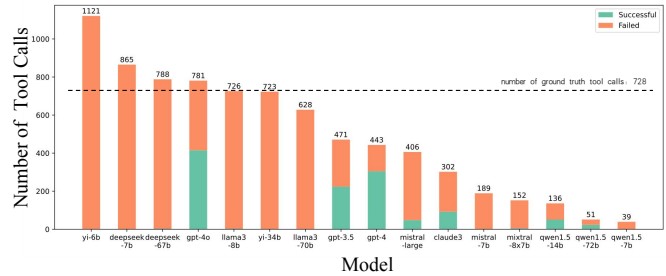

Figure 6: The number of successful and failed tool calls of each model.

## 4.3 Main Results

**Real-world tool-use tasks are challenging for existing LLMs.** Current LLMs are struggling to accurately invoke tools to solve these real-world tasks. As shown in Table 4, the best-performing models, GPT-4 and GPT-4o can only correctly solve fewer than 50% of the problems, while the rest of the models solve less than 25%. This shows that real-world problems with implicit steps, real tool invocations, and multimodal contextual inputs impose high requirements on the tool-use capabilities of LLMs. Regarding model performance comparisons, API-based models outperform open-source ones. Among open-source models, Qwen1.5-72B-Chat has the highest result accuracy. Larger models within the same series perform better than their smaller counterparts, but larger models from different series do not necessarily outperform the smaller ones, as shown in Figure 4. For example, the AnsAcc of Llama-3-70B-Instruct is higher than that of Llama-3-8B-Instruct, but lower than Qwen1.5-7B-Chat.

**The current bottleneck mainly lies on argument prediction.** From the results, we observe that the overall performance of the system is affected by the lowest metric. We argue that *the four metrics in the step-by-step mode follow the buckets effect*. To verify this observation, we calculate the Pearson correlation coefficients between four metrics (InstAcc, ToolAcc, ArgAcc, SummAcc) and AnsAcc, the result is shown in Figure 5. We find that the correlation coefficient for ArgAcc with AnsAcc is the highest. ArgAcc is low for most models, indicating that the four metrics follow the buckets effect. For example, the scores of Llama-3-70B-Instruct in InstAcc, ToolAcc, and SummAcc are higher than those of Qwen1.5-14B-Chat, but its ArgAcc is lower than Qwen1.5-14B-Chat, resulting in a lower final answer accuracy. The scores of GPT-4o in InstAcc and ToolAcc are higher than GPT-4, but its weaker argument prediction capability leads to a lower accuracy rate in the final result. The reason for the buckets effect is that under our evaluation framework, the model needs to follow user instructions, invoke tools multiple times in the correct format, and summarize the answer based on the returned results. Any error in this process can lead to an incorrect conclusion. Currently, argument prediction is the weakest capability for most models, suggesting that to enhance their general tool-use capabilities, researchers can focus on argument prediction capabilities. This concerns both the value and the format correctness of an argument.

**Different series of LLMs exhibit distinct behavioral patterns.** We count the number of successful and failed tool calls, illustrated in Figure 6. Successful means there are not any errors in the tool call. GPT-4o has the highest number of successful tool calls, while GPT-4 has the highest successful tool call rate. We find that models from different series exhibit dis-

Table 6: Detailed error distribution of GPT-4-1106-Preview and Llama-3-8B-Instruct on argument prediction.

| Error Type | GPT-4-1106-Preview | Llama-3-8B-Instruct |
|---|---|---|
| **Planning error:** | | |
| No action, requesting more information from the user. | 24 (38.7%) | 0 |
| No action, the whole response is model thought. | 31 (50%) | 2 (0.1%) |
| **Format error:** | | |
| The arguments do not follow the correct JSON format. | 5 (8.1%) | 118 (45.4%) |
| Trying to call multiple tools in one step. | 1 (1.6%) | 43 (16.5%) |
| Generating redundant information that leads to incorrect argument parsing. | 0 | 51 (19.6%) |
| Repeating contents from the prompt. | 0 | 41 (15.8%) |
| Generating the final answer but not following the correct format. | 1 (1.6%) | 5 (1.9%) |
| **Total number** | 62 (100%) | 260 (100%) |

tinct behavioral tendencies. Yi and Deepseek series tend to be *aggressive*, invoking tools frequently but lacks sufficient instruction-following ability to invoke tools in a correct format. The Qwen series is *conservative*, preferring to invoke tools less often, yet it has stronger instruction-following capabilities, resulting in a higher success rate of tool calls. The GPT series is *neutral*, tending to invoke tools moderately and possessing robust instruction-following abilities, which leads to the highest final answer accuracy. This suggests that to improve the performance of Yi or Deepseek, focus should be given to enhancing their instruction-following ability. Conversely, to enhance the Qwen series, reducing its conservative behavior to tool calls could be beneficial.

**Models favor either format errors or argument format errors, not both equally.** We count the percentage of error types when calling tools, including format error, argument format error, and N/A (other errors, mainly containing the tools' internal error). Most models exhibit a clear tendency toward either format errors or argument format errors, rather than making both types of mistakes in nearly equal numbers. For example, Claude-3's errors are predominantly argument format-related, amounting to 82.86%, while format errors account for a mere 4.29%. This indicates that Claude-3 can follow the tool-call format well, but fails to pass the argument in a correct format.

Table 5: The percentage of different error types.

| Model | Format Error (%) | Arg. Format Error (%) | N/A (%) |
|---|---|---|---|
| GPT-3.5-Turbo | 8.1 | 60.32 | 20.24 |
| GPT-4-1106-Preview | 70.29 | 4.35 | 25.36 |
| GPT-4o | 78.69 | 19.13 | 13.39 |
| Claude-3-Opus | 4.29 | 82.86 | 4.29 |
| Mistral-Large | 4.47 | 72.07 | 3.07 |
| Llama-3-8B-Instruct | 20.47 | 65.15 | 14.38 |
| Llama-3-70B-Instruct | 29.51 | 69.7 | 0.8 |
| Mistral-7B-Instruct | 49.21 | 46.56 | 4.23 |
| Mixtral-8x7B-Instruct | 53.74 | 40.82 | 5.44 |
| Qwen1.5-7B-Chat | 2.56 | 89.74 | 7.69 |
| Qwen1.5-14B-Chat | 2.35 | 71.76 | 25.88 |
| Qwen1.5-72B-Chat | 10.71 | 71.43 | 17.86 |
| Yi-6B-Chat | 98.22 | 0.18 | 1.61 |
| Yi-34B-Chat | 88.11 | 6.22 | 5.67 |
| Deepseek-LLM-7B-Chat | 52.49 | 19.65 | 27.86 |
| Deepseek-LLM-67B-Chat | 58.22 | 34.39 | 7.39 |

### 4.4 Further Analysis and Exploration

**Detailed Error Analysis.** In Section 4.3, we discuss the bottleneck in task performance arising from most models' inability to generate responses or predict arguments in the correct format. To understand the reason behind these model failures, we conduct a detailed analysis of the predictions generated by GPT-4-1106-Preview and Llama-3-8B-Instruct. We systematically categorize seven primary error types. The statistical outcomes are presented in Table 6. Detailed error cases of each type can be found in Appendix D.4.

Our analysis reveals distinct error distributions between GPT-4 and Llama-3. GPT-4 consistently adheres to the given prompts when executing actions, in contrast to Llama-3, which often fails to maintain the prescribed format. However, GPT-4 is prone to generating passive thought processes or attempting interaction with the user, rather than taking decisive action.

For the GPT-4 model, the predominant type of error is No Action, wherein the model neither utilizes tools nor produces a final answer. In 38.7% of erroneous responses, GPT-4 attempts to engage with

Table 7: Comparison of Llama-2-Chat-7B with Agent-Flan-7B (which is fine-tuned from Llama-2-Chat-7B on ReAct and JSON format data) on GTA.

| Model | Inst. | Tool. | Arg. | Summ. | Ans. | Ans.+I |
|---|---|---|---|---|---|---|
| Llama-2-Chat-7B | 30.86 | 16.34 | 0.36 | 47.69 | 3.96 | 2.98 |
| Agent-Flan-7B | **71.60** | **41.11** | **6.82** | **52.88** | **6.45** | **4.85** |

the user, mistakenly assuming the query lacks clarity and requesting additional information, despite the query and input images supplying sufficient details for task resolution. Furthermore, 50% of the error responses consist solely of the model's internal thought without any corresponding action.

For the Llama-3 model, most errors are related to formatting during action sequences, such as invoking tools or generating the final answer. Specifically, 45.4% of errors originate from argument predictions not adhering to a valid JSON format. Additionally, in 16.5% of the flawed responses, the model attempts to invoke multiple tools simultaneously, which is not supported by the agent system. Moreover, 19.6% of the errors occur when the model disregards the prompt and generates redundant information after argument prediction, leading to incorrect argument parsing. Finally, in 15.8% of the cases, the model fails to perform the correct action, merely repeating content from the prompt.

**Further Exploration to Enhance Model Performance.** Since the LLM functions as the central controller of an agent system, producing responses that strictly comply with the agent protocol is important. Fine-tuning on ReAct and JSON format may mitigate format-related errors during action execution. To verify this, we further compare Llama-2-Chat-7B with Agent-Flan-7B on GTA benchmark. AgentFLAN[6] is a popular instruction tuning method that fine-tunes LLM-based agents using ReAct and JSON instruction-following data. Agent-Flan-7B is fine-tuned from Llama-2-Chat-7B using AgentFLAN method. The results are shown in Table 7.

We discovered that Agent-Flan-7B's InstAcc and ToolAcc metrics were significantly higher than those of Llama-2-Chat-7B. The responses of Agent-Flan-7B follow the format of 'Thought-Action-Action Input' that specified in the prompt. But most responses of Llama-2-Chat-7B fail to follow the format. This further suggests the improved instruction following capability of Agent-Flan-7B.

However, we note that the ArgAcc of Agent-Flan-7B is still low. We compare the response of the two models, as shown in Appendix D.5. We find that although Agent-Flan-7B follows the format of 'Thought-Action-Action Input', it sometimes fails to generate the argument (Action Input) in a correct JSON format, or summarizes the final answer incorrectly. Thus, how to further enhance the model's capabilities on GTA through instruction fine-tuning is still an open problem.

## 5 Conclusion

We propose GTA, a real-world tool-use benchmark for general-purpose agents. The user queries are human-designed, step-implicit, and settled in real-world scenarios. Multimodal contextual inputs are provided. We build an evaluation platform equipped with executable tools in the categories of perception, operation, logic, and creation. Fine-grained metrics are designed for the tool-use capabilities of LLMs in real-world scenarios. We evaluate the tool-use capabilities of 16 LLMs. The evaluation results show that GTA is challenging for current LLMs, with advanced models like GPT-4 struggling with these real-world tasks, completing less than 50% of them. Based on our findings, we give takeaways and further suggestions on tool-use capability improvement. We believe that the GTA benchmark will advance further research in identifying the model's tool-use capabilities and contribute to realizing general-purpose tool agents.

## 6 Limitations

Our benchmark lacks language diversity since all queries are in English. Multilingual queries can be added in future work to assess the capability of tool agents in non-English environments. Moreover, to achieve high data quality, both the user queries and the tool chains are human-written. So the cost of a data piece is higher than that of AI-generated counterparts.

# 7 Acknowledgements

This work is supported by the National Key R&D Program of China (No. 2022ZD0161600), and the National Natural Science Foundation of China under Grants 62422311 and 62176152.

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

## A  Datasheet for Datasets

### A.1  Motivation

- **For what purpose was the dataset created?**

  We create GTA (a benchmark for General Tool Agents) to evaluate the general tool-use ability of LLMs in real-world scenarios. The benchmark has human-written queries with simple real-world objectives but implicit tool-use, an evaluation platform equipped with executable tools across diverse categories, and authentic image files as context input. These features bridge the gap between existing benchmarks and real-world tool-use scenarios.

- **Who created the dataset (e.g., which team, research group) and on behalf of which entity (e.g., company, institution, organization)?**

  The authors of this paper.

- **Who funded the creation of the dataset?**

  This work is supported by the National Key R&D Program of China (No. 2022ZD0161600), and the National Natural Science Foundation of China under Grants 62422311 and 62176152.

### A.2  Composition

- **What do the instances that comprise the dataset represent (e.g., documents, photos, people, countries)?**

  Each instance in GTA is in the JSON format. It contains natural language queries, image file inputs, tool descriptions, a reference tool chain, and a final answer.

- **How many instances are there in total (of each type, if appropriate)?**

  There are 229 instances in GTA, with 252 image files.

- **Does the dataset contain all possible instances or is it a sample (not necessarily random) of instances from a larger set?**

  We will provide all instances in our GitHub repository for GTA.

- **What data does each instance consist of?**

  Each instance contains a natural language query, image file inputs, tool descriptions, a reference tool chain, and a final answer.

- **Is there a label or target associated with each instance?**

  The correct tool chain and final answer is provided for each query.

- **Is any information missing from individual instances?**

  No.

- **Are relationships between individual instances made explicit (e.g., users' movie ratings, social network links)?**

  No.

- **Are there recommended data splits (e.g., training, development/validation, testing)?**

  The whole dataset is a test set.

- **Are there any errors, sources of noise, or redundancies in the dataset?**

  The dataset are created and verified by human. The noise may come from human error in writing.

- **Is the dataset self-contained, or does it link to or otherwise rely on external resources (e.g., websites, tweets, other datasets)?**

  The dataset is self-contained.

- **Does the dataset contain data that might be considered confidential (e.g., data that is protected by legal privilege or by doctor–patient confidentiality, data that includes the content of individuals' non-public communications)?**
  No.

- **Does the dataset contain data that, if viewed directly, might be offensive, insulting, threatening, or might otherwise cause anxiety?**
  No.

### A.3 Collection Process

- **How was the data associated with each instance acquired?**
  The queries are all human designed. The image inputs are collected from the Internet or created by annotators (such as diagrams drawn by annotators).

- **What mechanisms or procedures were used to collect the data (e.g., hardware apparatuses or sensors, manual human curation, software programs, software APIs)?**
  We use Google Images to collect image inputs. Queries are written by human.

- **Who was involved in the data collection process (e.g., students, crowdworkers, contractors) and how were they compensated (e.g., how much were crowdworkers paid)?** The data are created by researchers and student annotators. The annotators were paid about $ 40 per day.

- **Over what timeframe was the data collected?**
  The data were constructed in 2023 and 2024.

- **Were any ethical review processes conducted (e.g., by an institutional review board)?**
  Yes. All images within GTA are available for academic use. During the collection process, we instruct annotators to document the original URL of each image. Subsequently, we manually review these URLs, eliminating images that are not suitable for academic use. Moreover, should any authors request the removal of their images from GTA, we will promptly comply.

### A.4 Preprocessing/cleaning/labeling

- **Was any preprocessing/cleaning/labeling of the data done (e.g., discretization or bucketing, tokenization, part-of-speech tagging, SIFT feature extraction, removal of instances, processing of missing values)?**
  The dataset is created by human from scratch, and verified manually.

- **Was the "raw" data saved in addition to the preprocessed/cleaned/labeled data (e.g., to support unanticipated future uses)?**
  There is no raw data, since the dataset is created from scratch, rather than a cleaned version of existing data.

- **Is the software that was used to preprocess/clean/label the data available?**
  Excel and VSCode are used for create the data.

### A.5 Uses

- **Has the dataset been used for any tasks already?**
  No.

- **Is there a repository that links to any or all papers or systems that use the dataset?**
  No.

- **What (other) tasks could the dataset be used for?**
  GTA is used for evaluating the general tool-use ability of LLMs in real-world scenarios.

- **Is there anything about the composition of the dataset or the way it was collected and preprocessed/cleaned/labeled that might impact future uses?**
  No.

- **Are there tasks for which the dataset should not be used?**

  No.

- **Are there any potential negative social impacts?**

  The GTA benchmark may have potential negative societal impacts. These include copyright concerns related to image data collection. The presence of images involving people in our dataset also raises privacy concerns. Additionally, during the evaluation of GTA, the agent system could potentially experience hallucinations and generate harmful information. Besides, given the inclusion of coding questions in GTA, the agent system might produce malicious code.

## A.6 Distribution

- **Will the dataset be distributed to third parties outside of the entity (e.g., company, institution, organization) on behalf of which the dataset was created?**

  No.

- **How will the dataset will be distributed (e.g., tarball on website, API, GitHub)?**

  The dataset will be released at https://github.com/open-compass/GTA.

- **Will the dataset be distributed under a copyright or other intellectual property (IP) license, and/or under applicable terms of use (ToU)?**

  The dataset is released under the Apache License.

- **Have any third parties imposed IP-based or other restrictions on the data associated with the instances?**

  No.

- **Do any export controls or other regulatory restrictions apply to the dataset or to individual instances?**

  No.

## A.7 Maintenance

- **Who will be supporting/hosting/maintaining the dataset?**

  The authors of this paper.

- **How can the owner/curator/manager of the dataset be contacted (e.g., email address)?**

  Please contact with authors through emails in the paper.

- **Is there an erratum?**

  No.

- **Will the dataset be updated (e.g., to correct labeling errors, add new instances, delete instances)?**

  Yes, users can propose issues and the dataset will be updated on Github.

- **Will older versions of the dataset continue to be supported/hosted/maintained?**

  Primarily, we plan to maintain only the most recent version of the dataset. However, under certain circumstances, such as significant updates to our dataset or the need for validation of previous research work using older versions, we will exceptionally preserve previous versions of the dataset for up to one year.

- **If others want to extend/augment/build on/contribute to the dataset, is there a mechanism for them to do so?**

  Contact the authors of the paper.

# B  Additional Information of GTA

## B.1  Tool Definition

The detailed definition of 14 tools across perception, operation, logic, and creativity categories are shown in Table 8.

Table 8: Detailed definition of 14 tools across four categories.

| Name | Description | Input | Output |
|------|-------------|-------|--------|
| *- Perception* | | | |
| OCR | Recognize the text from an image. | [image] An image containing text. | [text] The text on the image. |
| RegionAttributeDesc. | Describe a certain attribute of a certain part in the input image. | [image] Any image. [text] Region location and the name of attribute to describe. | [text] The description of the region. |
| DetectGivenObject | Detect certain object in the image. | [image] Any image. [text] Object name. | [image] An image with bounding box. [text] The location of bounding box and detecting scores. |
| ImageDescription | Describe the input image. | [image] Any image. | [text] The description of the image. |
| *- Operation* | | | |
| DrawBox | Draw a box on a certain location of the image. | [image] Any image. [Text] Box location. | [image] An image with a box on the certain location. |
| AddText | Add text on the image. | [image] Any image. [Text] Text, font size, and location. | [image] An image with text on the certain location. |
| GoogleSearch | Search on Google. | [text] The content to search. | [text] Searching results. |
| *- Logic* | | | |
| Calculator | Calculate by Python interpreter. | [text] Math expressions including only numbers and operation symbols. | [text] Calculation result. |
| Plot | Use code interpreter to draw math diagrams, statistics, etc. | [text] Python codes using Matplotlib to draw a diagram. | [image] The diagram. |
| MathOCR | Recognize the math expressions from a image. | [image] An image containing math expression. | [text] Latex format of the math expression. |
| CountGivenObject | Count the number of certain objects in the image. | [image] Any image. [text] The object name. | [text] The number of the object contained in the image. |
| Solver | Use code interpreter to solve math expressions. | [text] Python codes using Sympy to solve math equations or expressions containing unknown variables. | [text] Solving results. |
| *- Creativity* | | | |
| TextToImage | Generate an image from the input text. | [text] The description of an image. | [image] The image generated. |
| ImageStylization | Transfer the style of the image as that of a reference image. | [text] The description of the target image style. [image] An image to be transferred. | [image] The target image in the style of the text description. |

## B.2  Examples of Three Query Types

The examples of objective queries $\mathcal{Q}_o$, subjective queries $\mathcal{Q}_s$, and image generation queries $\mathcal{Q}_g$ are shown in Figure 7, Figure 8, and Figure 9, respectively. In the supplementary material, we provide the complete data sample, which is in the JSON format, including the involved tools, files, query, tool chain, and the final answer. To facilitate automatic evaluation, we design different final answer format for the three query types. For objective queries, the final answer contains both a whitelist and a blacklist of phrases. An answer is considered correct if it includes all terms from the whitelist and excludes all terms from the blacklist. In the case of subjective queries, the final answer contains three manually labeled responses from distinct annotators. We compute the cosine similarity (ranging from 0 to 1) between the model's prediction and each of the three ground truth answers, ultimately considering the highest score obtained. For image generation queries, the final answer is none, since

we evaluate the execution accuracy through measuring the argument accuracy of image generation tools.

---

**Query Type:** Objective
**Query:** I need to prepare twelve servings of this dish. How many boxes of eggs will I need in total?
**Involved Tools:** ImageDescription, CountGivenObject, OCR
**Files:**

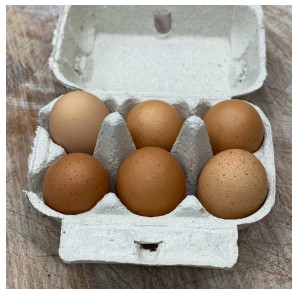

**Steps:**

1. Count the number of eggs in the photo.
2. Identify the eggs needed for one serving of a dish on the recipe.
3. Calculate how many eggs are needed for 12 dishes.
4. Calculate how many boxes of eggs are needed.

**Answer:** 2

---

Figure 7: An example of objective query $\mathcal{Q}_o$. The final answer is a uniquely determined number or phrase.

---

**Query Type:** Subjective
**Query:** According to the sign, what should I avoid to do now? Why?
**Involved Tools:** ImageDescription, OCR
**Files:**

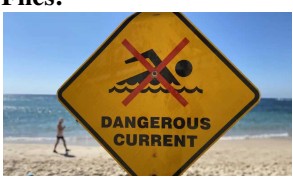

**Steps:**

1. Recognize the image background and the icon on the sign.
2. Recognize the text in the picture.

**Answer:** You should avoid swimming due to the dangerous current.

---

Figure 8: An example of subjective query $\mathcal{Q}_s$. The final answer is usually some descriptive text. It is not unique, but the general idea is the same.

## C Additional Information for Data Design

### C.1 Query Exemplars

We design several initial queries as query exemplars, as shown from Figure 10 to 24. The annotators brainstorm and design new questions that have the same tool chain as the exemplar but with different scenarios. We provide an expansion example for most exemplars for annotators to refer to.

**Query Type:** Image Generation
**Query:** I want to go to the highest-rated restaurant. Please circle it in the map.
**Involved Tools:** OCR, DrawBox

**Files:**                                                    **Generated Image:**

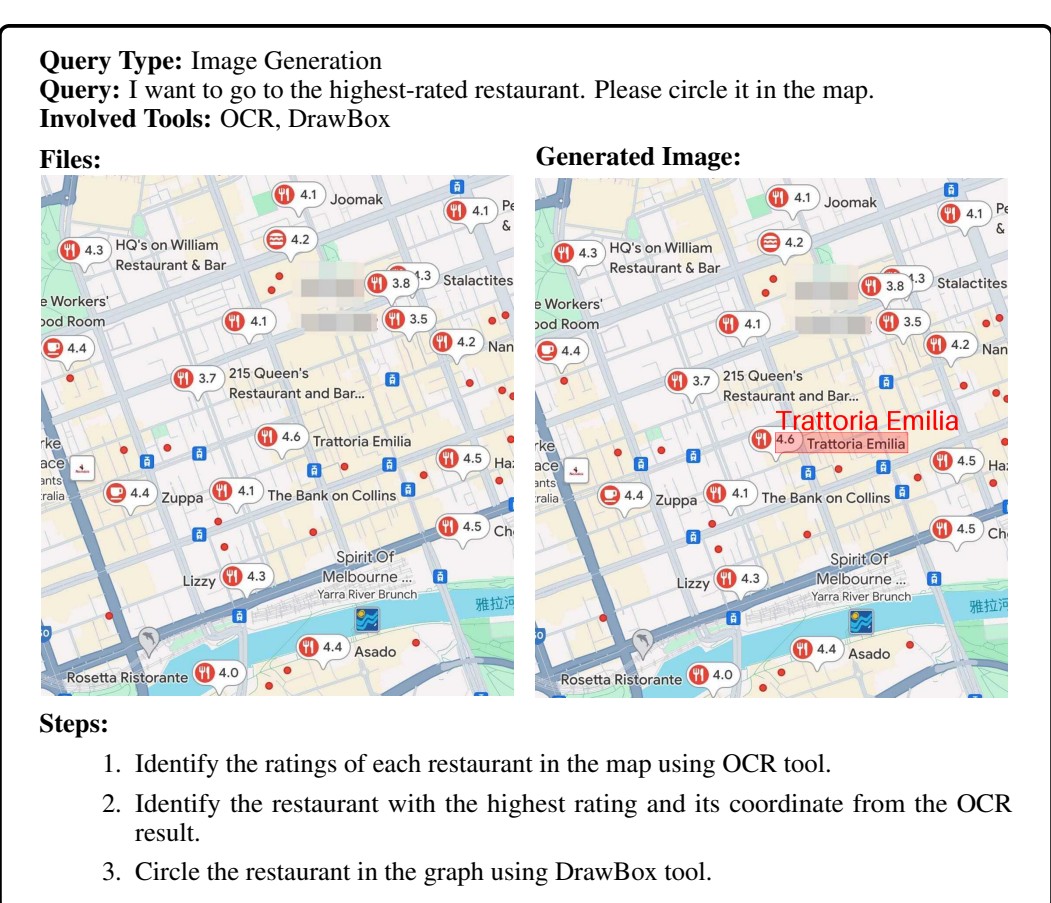

**Steps:**

1. Identify the ratings of each restaurant in the map using OCR tool.

2. Identify the restaurant with the highest rating and its coordinate from the OCR result.

3. Circle the restaurant in the graph using DrawBox tool.

Figure 9: An example of image generation query $\mathcal{Q}_i$. The final answer is none since we do not evaluate the generated image directly.

## Exemplar 1

**Query:** How much should I pay for the beer on the table according to the price on the menu?
**Involved Tools:** ImageDescription, CountGivenObject, OCR, Calculator
**Files:**

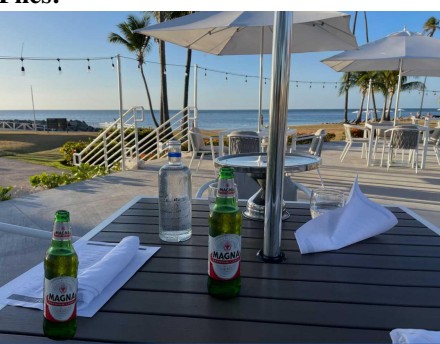 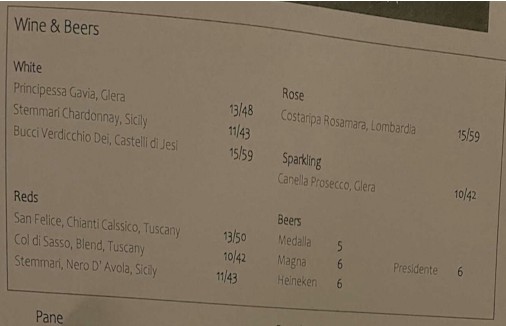

**Steps:**

1. Count the number of beers.
2. Recognize text on the bottles.
3. Recognize text on the menu.
4. Calculate the total price of the beers.

**Answer:** 12

## Expansion Example

**Query:** I need to prepare twelve servings of this dish. How many boxes of eggs will I need in total?
**Involved Tools:** ImageDescription, CountGivenObject, OCR, Calculator
**Files:**

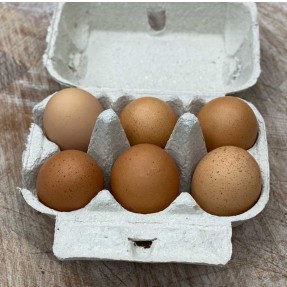

**Steps:**

1. Count the number of eggs in the photo.
2. Identify the eggs needed for one serving of a dish on the recipe.
3. Calculate how many eggs are needed for 12 dishes.
4. Calculate how many boxes of eggs are needed.

**Answer:** 2

Figure 10: Query exemplar 1.

**Exemplar 2**

**Query:** Can you explain this meme?
**Involved Tools:** OCR, ImageDescription

**Files:**

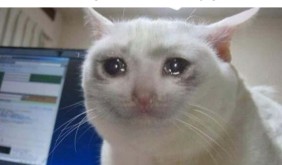

**Steps:**

1. Recognize the text in the picture.

2. Describe the content of the image.

3. Infer the central idea in relation to the image and the text.

**Answer:** The meme shows it is sad when we send a message to a friend who's online and right after that, they go offline. It's a coincidental and unpleasant situation.

**Expansion Example**

**Query:** What sports event was this photo taken at? Please provide the names of the two opposing teams in your answer.
**Involved Tools:** OCR, ImageDescription

**Files:**

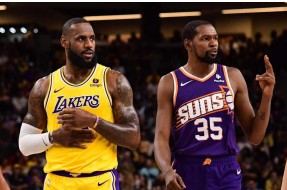

**Steps:**

1. Identify the words in the picture: Lakers, Suns.

2. Describe the content of the picture: basketball game.

**Answer:** Lakers vs suns basketball game.

Figure 11: Query exemplar 2.

**Exemplar 3**

**Query:** What is the woman in a pink shirt doing?
**Involved Tools:** DetectGivenObject, RegionAttributeDescription
**Files:**

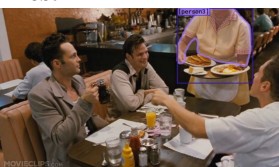

**Steps:**

1. Detect the woman in pink.

2. Describe the action of the person in the detection box.

**Answer:** Serving food.

**Expansion Example**

**Query:** What is the breed of the dog in the middle of the picture?
**Involved Tools:** DetectGivenObject, RegionAttributeDescription

**Files:**

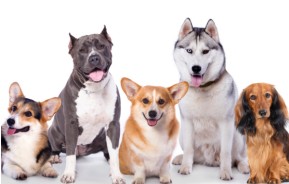

**Steps:**

1. Detect all the dogs.

2. Find the detection box in the center.

3. Describe the dog's breed in the detection box.

**Answer:** Corgi.

Figure 12: Query exemplar 3.

## Exemplar 4

**Query:** What is x in the equation?
**Involved Tools:** MathOCR, Solver

**Files:**                    **Steps:**

$$(x+3)^2=4$$

1. Convert the handwritten image into latex style.
2. Solve the equation.

**Answer:** -1 or -5.

### Expansion Example

**Query:** What is the image of this analytic formula?
**Involved Tools:** MathOCR, Plot

**Files:**                    **Steps:**

$$y = x^2 + 2x - 1$$

1. Convert the handwritten image into latex style.
2. Plot according to the math expression.

Figure 13: Query exemplar 4.

## Exemplar 5

**Query:** Convert the table into a statistical chart with the type of image shown in the example. The horizontal axis is the country, and the vertical axis uses three colors for sales volume, revenue, and profit.
**Involved Tools:** ImageDescription, OCR, Plot

**Files:**

| Country | Sales Volume | Revenue | Profit |
|---|---|---|---|
| USA | 40.080 | $15.971.880 | $3.086.421 |
| China | 35.070 | $15.866.670 | $3.032.162 |
| Australia | 27.054 | $14.812.566 | $2.868.636 |
| India | 23.046 | $10.608.174 | $1.853.710 |
| South Korea | 16.032 | $10.494.948 | $1.975.844 |

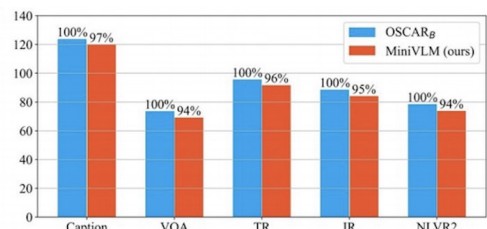

**Steps:**

1. Recognize text in the table.
2. Describe the style of the statistical chart.
3. Plot the diagram in the same style with the data from the table.

Figure 14: Query exemplar 5.

## Exemplar 6

**Query:** What percentage of people wear helmets?
**Involved Tools:** DetectGivenObject, RegionAttributeDescription, Calculator
**Files:**

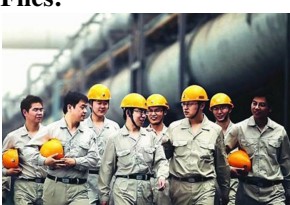

**Steps:**

1. Detect all the people.
2. Describe each of the people whether he wears a helmet.
3. Calculate the percentage.

**Answer:** 62.5%.

### Expansion Example

**Query:** What's the total number of the mother swans and the baby swans?
**Involved Tools:** CountGivenObject, ImageDescription, Calculator

**Files:**

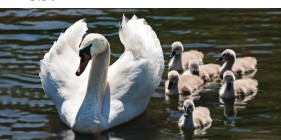

**Steps:**

1. Count the number of mother swans.
2. Count the number of baby swans.
2. Calculate the total number.

**Answer:** 7.

Figure 15: Query exemplar 6.

## Exemplar 7

**Query:** I'm a 23-year-old female. How many grams of this kind fruit can I meet the vitamin C intake recommended by U.S. Recommended Dietary Allowance in 2021? Please round your answers to the nearest gram. You can look for information in National Institutes of Health and Wikipedia.

**Involved Tools:** ImageDescription, GoogleSearch, Calculator

### Steps:

**Files:**

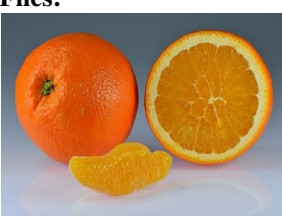

1. Identify the fruit in the picture as an orange.

2. Search Wikipedia for the VC content of oranges: 53mg/100g.

3. Search National Institutes of Health's recommended VC intake for adults: 75mg for women, 90mg for men.

4. Calculate the intake of oranges = recommended VC intake (I'm a woman, take 75mg)/VC content, and round it up.

**Answer:** 142.

**Evidence:**

https://en.wikipedia.org/wiki/Vitamin_C
https://ods.od.nih.gov/factsheets/VitaminC-HealthProfessional/

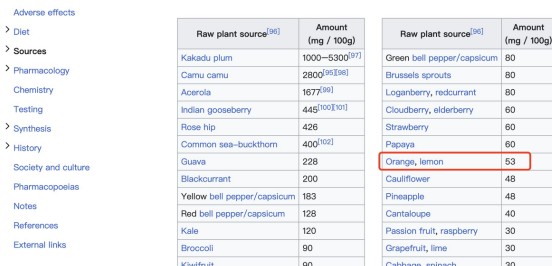
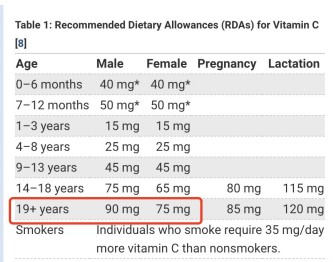

## Expansion Example

**Query:** According to Midwest Dairy, how many gallons of milk can this animal produce at most in 725 days?

**Involved Tools:** ImageDescription, GoogleSearch, Calculator

**Files:**

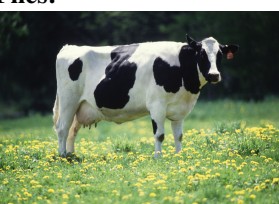

### Steps:

1. Identify the animal in the image as a dairy cow.

2. Search for the average daily milk production for cows recorded on Midwest Dairy: 6-7 gallons.

3. Calculate the maximum production over a 725 day period: 725*7.

**Answer:** 5075.

**Evidence:**

https://www.midwestdairy.com/farm-life/farm-life-faq/

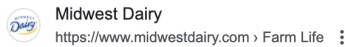

Most dairy cows are milked two to three times per day. On average, a cow will produce six to seven gallons of milk each day.

Midwest Dairy
https://www.midwestdairy.com › Farm Life
Farm Life FAQ - Midwest Dairy

Figure 16: Query exemplar 7.



**Exemplar 8**

**Query:** How much did I spend on food totally?
**Involved Tools:** OCR, Calculator
**Files:**

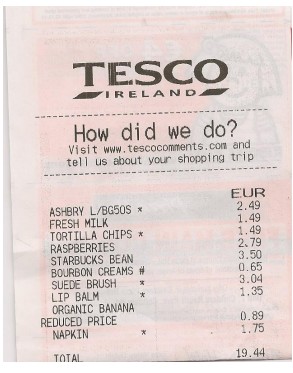

**Steps:**

1. Identify goods and their prices.

2. Identify the food in the bill.

3. Calculate the total price of the food.

**Answer:** 10.81

**Expansion Example**

**Query:** We are a family of 5 and everyone takes fish oil. How many days is this bottle of fish oil enough for us?
**Involved Tools:** OCR, Calculator

**Files:**

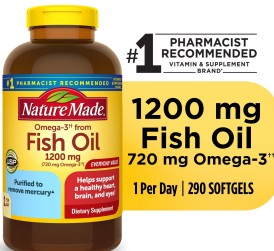

**Steps:**

1. Identify key information from the bottle: 1 per day, 290 softgels.

2. Calculate the bottle number: 290/5.

**Answer:** 58



Figure 17: Query exemplar 8.

**Exemplar 9**

**Query:** I have 22 dollars. For lunch, my mom and I would each like an entree and a dessert. I don't eat doughnuts and my mom doesn't eat chicken. All of our food should be different. What specific foods can I buy?

**Involved Tools:** OCR, Calculator

**Files:**

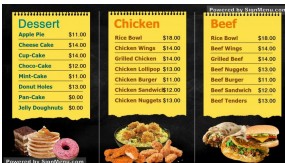

**Steps:**

1. Identify dishes and prices.
2. Find the food that meets the constraints.
3. Find out the food with total price less than $22.

**Answer:** For you, a Chicken Burger for the entree and a Pan-Cake for the dessert. For your mom, a Beef Burger for the entree and a Jelly Doughnuts for the dessert.

**Expansion Example**

**Query:** I need a total ethereum hash rate of at least 122 MH/s, and the total rated power should not exceed 510 W. Which two GPU should I buy?

**Involved Tools:** OCR, Calculator

**Files:**

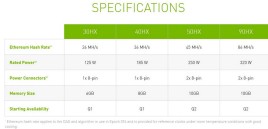

**Steps:**

1. Identify GPUs and their prices.
2. Find out GPUs with summed power greater than 122MH/s and less than 510W.

**Answer:** One 40HX and one 90HX.

Figure 18: Query exemplar 9.

**Exemplar 10**

**Query:** I want to make this dish. How many grams of pork mince do I need according to BBC Good Food?

**Involved Tools:** ImageDescription, GoogleSearch

**Files:**

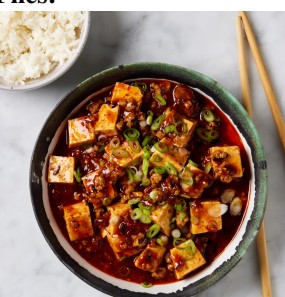

**Steps:**

1. Identify the dish.
2. Search BBC Good Food for recipes and ingredient lists.
3. Find out the gram number of pork mince.

**Answer:** 100

**Evidence:**

https://www.bbcgoodfood.com/recipes/mapo-tofu

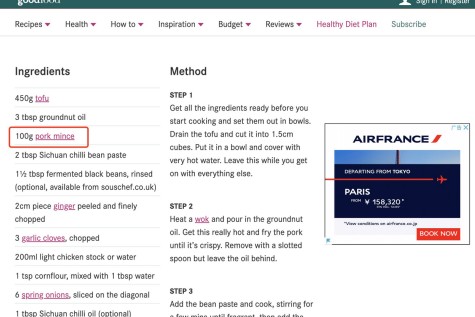

**Expansion Example**

**Query:** I want to go to this place in Shanghai, place tell me it's "Regular" ticket price in June, 2023. Please answer in RMB.

**Involved Tools:** ImageDescription, GoogleSearch

**Files:**

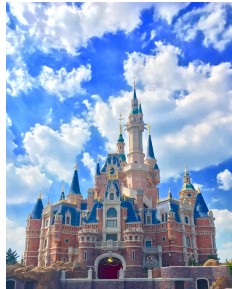

**Steps:**

1. Identify the building in the picture.
2. Search for "Regular" ticket price for Shanghai Disney in 2023.

**Answer:** 475

**Evidence:**

https://www.shanghaidisneyresort.com/en/new-pricing-structure/

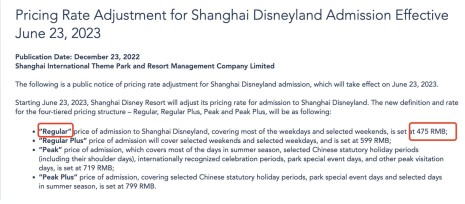

Figure 19: Query exemplar 10.

## Exemplar 11

**Query:** I will get off work at 5:00 today. I need to spend an hour for dinner and half an hour to get to the movie theater. Which is the earliest movie show I can catch? Please circle it in the screenshot.
**Involved Tools:** OCR, DrawBox
**Files:**

### MONDAY 1/15/24

| TIME | TITLE |
|------|-------|
| 5:00am | The Little Princess (1939) |
| *Featuring: Shirley Temple, Richard Greene* | |
| 7:00am | A Room With A View (1985) |
| *Featuring: Maggie Smith, Helena Bonham Carter* | |
| 9:35am | The Trip To Bountiful (1985) |
| *Featuring: Geraldine Page, John Heard* | |
| 11:55am | Cinderella Liberty (1973) |
| *Featuring: James Cann, Marsha Mason, Kirk Calloway* | |
| 2:25pm | Rough Magic (1995) |
| *Featuring: Bridget Fonda, Russell Crowe* | |
| 4:40pm | Friends with Kids (2011) |
| *Featuring: Adam Scott, Jennifer Westfeldt* | |
| 7:00pm | A Walk To Remember (2002) |
| *Featuring: Mandy Moore, Shane West* | |
| 9:10pm | If Only (2004) |
| *Featuring: Jennifer Love Hewitt, Paul Nicholls* | |
| 11:15pm | Across the Tracks (1990) |
| *Featuring: Brad Pitt, Ricky Schroder* | |
| 1:25am | Rock 'N' Roll High School (1979) |
| *Featuring: P.J. Soles, Vincent Van Patten* | |
| 3:25am | Detour (1945) |
| *Featuring: Tom Neal, Ann Savage* | |

**Steps:**

1. Calculate the arrival time at the movie theater.

2. Identify the start time of each movie.

3. Identify the earliest movie that is later than the arrival time.

4. Circle the movie in the image.

Figure 20: Query exemplar 11.

## Exemplar 12

**Query:** As of December 31, 2023, how many Boeing 787-8 Dreamliner airplanes does the airline shown in the image own?

**Involved Tools:** OCR, GoogleSearch

**Files:**

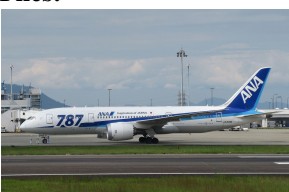

**Steps:**

1. Identify the airline name.
2. Search for the number of aircraft of the type owned by the airline company.

**Answer:** 36

**Evidence:**

https://en.wikipedia.org/wiki/All_Nippon_Airways

| Boeing 777–300 | 5 | — | — | — | 21 | 493 | 514 | To be retired. |
|---|---|---|---|---|---|---|---|---|
| Boeing 777–300ER | 13 | — | 8 | 68 64 | 24 | 112 116 | 212 | |
| Boeing 777–9 | — | 18 | | | TBA | | | To replace Boeing 777–300s and 13 older Boeing 777–300ERs.[77][78] Two aircraft were converted to Boeing 777–8F.[73] |
| Boeing 787–8 | 36 | — | — | 46 32 42 | 21 14 — | 102 138 198 | 169 184 240 | Launch customer. JA874A painted in "ANA Future Promise" livery.[79] |
| | | | | — | 12 | 323 | 335 | Equipped with domestic configuration. |
| Boeing 787–9 | 42 | 6 | — | 48 40 | 21 14 | 146 192 | 215 246 | JA873A painted in a Star Wars R2–D2 special livery. JA871A painted in "ANA Future Promise" livery Replacing older Boeing 777–200 and Boeing 777–300.[80] |
| | | | | 18 | 377 | 395 | | Equipped with domestic configuration. |

## Expansion Example

**Query:** How many cores does this cpu have?

**Involved Tools:** OCR, GoogleSearch

**Files:**

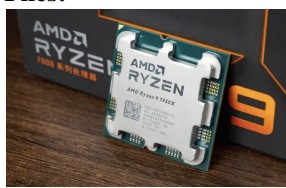

**Steps:**

1. Identify the CPU type.
2. Search for the core number of this CPU.

**Answer:** 16

**Evidence:**

https://www.amd.com/en/products/cpu/amd-ryzen-9-7950x

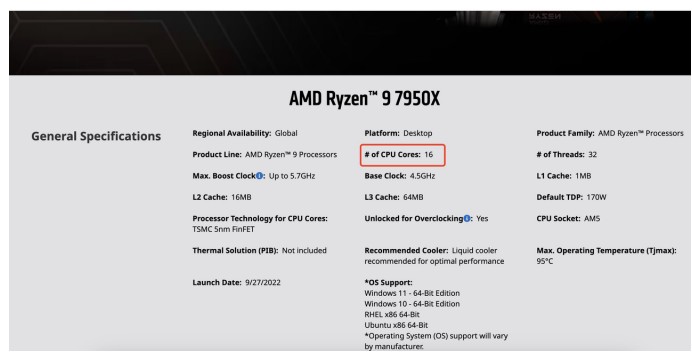

Figure 21: Query exemplar 12.

**Exemplar 13**

**Files:**

**Query:** This is part of a microwave oven control panel. I want to heat the food for 2 minutes. Which buttons should I press in sequence?
**Involved Tools:** OCR, Calculator
**Steps:**

1. Recognize button names.

2. Calculate the number of button presses according to heating time.

3. Plan the order of button presses.

**Answer:** 1 min button: once; 10 sec button: three times; the start button: once.

Figure 22: Query exemplar 13.

**Exemplar 14**

**Query:** Can you generate a picture of cake containing these ingredients?
**Involved Tools:** ImageDescription, TextToImage
**Files:**

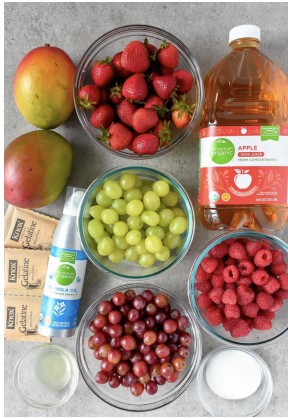

**Steps:**

1. Recognize the ingredients in the image.

2. Generate a picture of a cake containing these ingredients.

**Expansion Example**

**Query:** I want a picture of a boy walking on the grass. The boy is wearing a T-shirt in the same color as the girl's top in the picture.
**Involved Tools:** ImageDescription, TextToImage

**Files:**

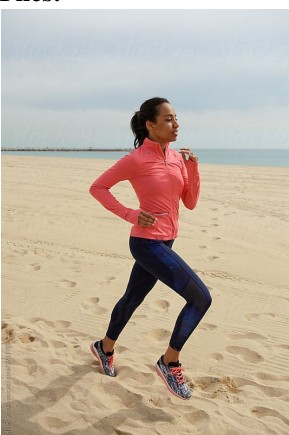

**Steps:**

1. Identify the girl's top color: pink.

2. Find the detection box in the center.

3. Generate a picture of a boy walking in the grass, the boy is wearing a pink t-shirt.

Figure 23: Query exemplar 14.

**Exemplar 15**

**Query:** Convert the photo to cartoon style. Generate a title and put it above the boy using font size 16.
**Involved Tools:** ImageStylization, ImageDescription, AddText, DetectGivenObject
**Files:**

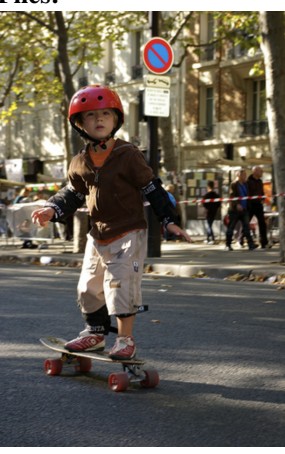

**Steps:**

1. Convert the image to cartoon style.

2. Describe the image and generate a caption.

3. Detect the position of the little boy.

4. Place the caption above the little boy using a font size of 16.

**Expansion Example**

**Query:** Make a short poem of 50 words or less based on the landscape in the picture. Convert the picture to an ink drawing and place the short poem in the upper right corner of the picture using font size 10.
**Involved Tools:** ImageStylization, ImageDescription, AddText, DetectGivenObject

**Files:**

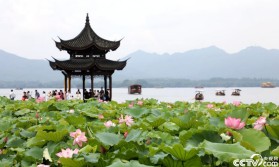

**Steps:**

1. Generate an image description and compose a poem based on the description.

2. Convert the image to ink painting style.

3. Put the text in the upper right corner of the generated picture.

Figure 24: Query exemplar 15.

## C.2 Diversified Expansion Approach

To ensure expansion diversity, we instruct annotators to design new questions according to the diversified expansion approach. Rules of the approach are shown in Figure 25. We also provide an example, shown in Figure 26.

> For each exemplar, adopt the three following approaches.
> **Approach One:** Keep the tools in the exemplar unchanged, change the question scenarios and design 6 new samples. These scenarios should be different from each other. An expansion example is provided for each exemplar.
> **Approach Two:** Replace one of the tools in the exemplar and design questions based on the new involved tool set. Design 2 new samples in this way.
> **Approach Three:** Increase or decrease the tools in the exemplar and design 2 new samples in this way according to the new involved tool set. The detailed rules are as follows:
>
>     i. If there are 2 tools in the exemplar: add 1 tool and design one sample; add 2 tools and design another sample.
>
>     ii. If there are 3 tools in the exemplar: reduce 1 tool and design one sample; increase 1 tool and design another sample.
>
>     iii. If there are 4 tools in the exemplar: reduce 1 tool and design one sample; reduce 2 tools and design another sample.

Figure 25: Diversified expansion approach.

**[Original Exemplar]**
**Query:** I'm a 23-year-old female. How many grams of this kind fruit can I meet the vitamin C intake recommended by U.S. Recommended Dietary Allowance in 2021? Please round your answers to the nearest gram. You can look for information in National Institutes of Health and Wikipedia.
**Involved Tools:** ImageDescription, GoogleSearch, Calculator

**Files:**

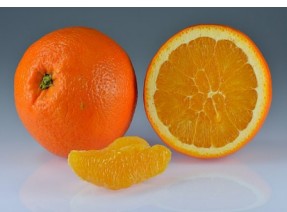

**Steps:**

1. Identify the fruit in the picture as an orange.
2. Search Wikipedia for the VC content of oranges.
3. Search National Institutes of Health's recommended VC intake for adults.
4. Calculate the intake of oranges = recommended VC intake/VC content, and round it up.

**Answer:** 142.

**[Approach One]**
**Query:** According to Midwest Dairy, how many gallons of milk can this animal produce at most in 725 days?
**Involved Tools:** ImageDescription, GoogleSearch, Calculator

**Files:**

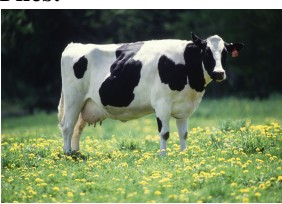

**Steps:**

1. Identify the animal in the image as a dairy cow.
2. Search for the average daily milk production for cows recorded on Midwest Dairy.
3. Calculate the maximum production over 725 days.

**Answer:** 5075.

**[Approach Two]**
**Query:** $0.80 for an apple, $1 for a pear, $0.90 for a banana. How many dollars do these fruits cost?
**Involved Tools:** ImageDescription, Calculator, CountGivenObject

**Files:**

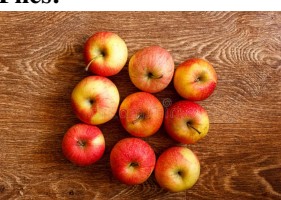

**Steps:**

1. Identify the fruit in the picture as apples.
2. Count the apples in the image.
3. Calculate the total price.

**Answer:** 7.2

**[Approach Three]**
**Query:** Assume that one bottle contains 500g drink, how many sugar does these drink contain? Please round your answers to the nearest gram. You can find information in USDA (U.S. Department of Agriculture).
**Involved Tools:** ImageDescription, Calculator, GoogleSearch, CountGivenObject

**Files:**

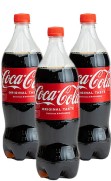

**Steps:**

1. Search for the sugar content of Coke in USDA.
2. Count the colas in the image.
3. Calculate the total sugar content.

**Answer:** 135

Figure 26: An example for the diversified expansion approach. Changes to the tool set are highlighted in blue. The evidence part is omitted for clarity of illustration.

## C.3 Instruction for Annotators

The detailed instruction for annotators during the query construction stage is provided in Figure 27. The instruction during the tool chain construction stage is provided in Figure 28.

---

**General Goal:**

- Design questions that require calling tools and go through multiple steps to solve. Each question should be based on one or two image files.
- We provide the tool list (B.1) and query exemplars (C.1). Please design more queries according to the rules described in the diversified expansion approach (C.2).

**Each sample should fulfill the following requirements:**

1. Each sample contains 6 parts: F (Image File), Q (Query), T (Tools), S (Steps), A (Answer), E (Evidence).

2. Image files can be sourced from the web and must be credited with a URL, or they can be created by the annotators themselves (e.g., through photography, drawing, etc.).

3. Q is the query posed based on the image. T is the tool needed to solve the problem. S is the steps to be taken to solve the problem. A is the answer to the question. The role of E is described in 8.

4. S needs to contain two or more steps.

5. Q needs to avoid obvious references to a tool (A counterexample: *Please detect the orange.* This statement clearly refers to the tool DetectGivenObject).

6. With regard to answer A, questions that generate text or images do not need to be answered, while the rest of the questions need to ensure that there is a single definitive answer and should not rely on images generated in previous steps. For example, the question *what kind of animal is in the picture* should not be asked after *generate an image of an animal*, as the answer is uncertain.

7. Q and A need to be in English. If there is text in the pictures, it can only be in English.

8. For questions that need the GoogleSearch tool, the URL and a screenshot containing the answer is required in E. Other questions are not required to provide E.

9. For questions that need the GoogleSearch tool, it is important to note that the question does need to be solved by searching (e.g., the question is time-sensitive, or it specifies which website to get the information from), rather than being potentially known by the LLM itself. (Counter example: *Tsinghua University is located in which city in China?* Positive example: *What is the QS ranking of Tsinghua University in 2023?* Counter example: *What is the recipe for Mapo Tofu?* Positive example: *What is the recipe for Mapo Tofu given on the BBC Good Food website?* Counter example: *How long is Trump's term in office?* Positive example: *According to Wikipedia, how long is Trump's term in office?*)

10. Questions that need the GoogleSearch tool are often time-sensitive. We need to ask them in a way that ensures the answers do not change over time. You should ensure that the question can be searched for a unique and definitive answer regardless of the time. To achieve this, you can specify the timeframe, webpage, organization, etc. to be searched for in your question. (Counter example: *What is the QS ranking of Tsinghua University?* Positive example: *What will be the QS ranking of Tsinghua University in 2023?*) Please record the URL and a screenshot containing the answer in E.

---

Figure 27: Annotation instruction document for query construction stage.

**General Goal:**

We have designed about 200 queries for LLM tool call evaluation. Now we would like to annotate a correct tool chain for each query. The deliverable is a JSON file.

**Each sample should fulfill the following requirements:**

1. To make it easier for you to annotate in the correct format, as shown in C.4, we generate a tool chain for each query using GPT-4 as an annotation example. Please annotate according to the format.

2. We have deployed all the tools. You should call the tools to solve the queries. You can refer to the S (Steps) recorded in the query file. Record the tool call argument and return value for each step.

3. Make sure that the tool always yields the correct answer for these queries. If the tool cannot recognize the image file correctly, just discard the query.

**How to call a tool:**

```
from agentlego.tools.remote import RemoteTool
tools = RemoteTool.from_server(server_url)

# Calculator
tools[0]('3+2')
# GoogleSearch
# arg2: number of results returned
tools[1]('Vitamin C content in oranges per 100g',4)
# OCR
tools[5]('image.jpg')
# ImageDescription
tools[6]('image.jpg')
# TextToBbox
# arg3:
# whether only return the bbox of the highest probability
tools[8]('image.jpg', 'apple', False)
# CountGivenObject
tools[9]('image.jpg', 'apple')
# MathOCR
tools[10]('image.jpg')
# DrawBox
tools[13]('image.jpg', '(49, 1, 342, 240)')
# TextToImage
tools[15]('man riding on the road')
# ImageStylization
tools[16]('image.jpg','convert to Picasso style')
```

Figure 28: Annotation instruction document for tool chain construction stage.

## C.4   Illustration of Executable Tool Chains

An illustration on each part of the tool chain is shown in Figure 29. It is in the JSON format. It contains the involved tool list, file list, and dialog list. There are three roles in the dialog list: user, assistant, and tool. In the user's dialog, the query content is recorded. In the assistant's dialog, the correct tool call including the tool name and arguments is recorded. In the tool's dialog, the tool's return value is recorded. You can refer to Figure 7 to ??, Figure 8 to ??, and Figure 9 to ?? for JSON-format tool chain examples.

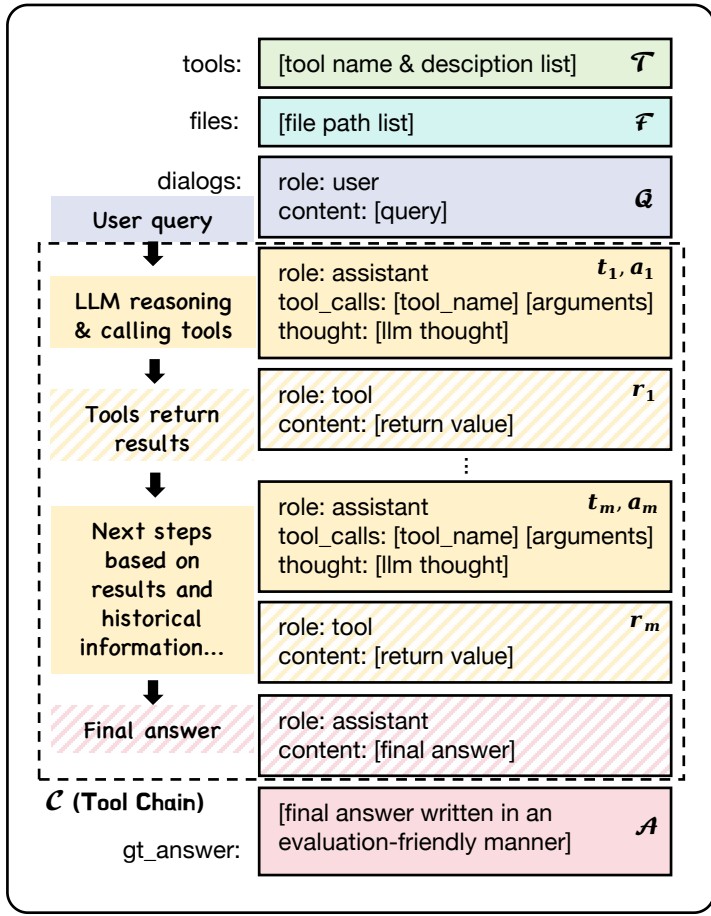

Figure 29: An illustration of each part of the tool chain.

# D  Additional Information for Experiments

## D.1  Build an LLM-Based Agent System

We build the LLM-based agent system using Lagent [2] framework. It equips an LLM with some action & planning schema, using action executor to let it interact with external tools. To build such an agent system, we should consider three parts: LLM, action & planning schema, and tools. In our experiment, we use ReAct as the action & planning schema. As for tools, we have implemented the 14 tools using AgentLego [3], which is a platform supporting tool serving and remote accessing. When evaluating different LLMs, we replace different LLMs into the Lagent framework, and evaluate this system on the Opencompass [4] evaluation platform.

## D.2  ReAct-Style Prompts

The ReAct-style prompt template using for the agent system is shown in Figure 30.

```
CALL_PROTOCOL_EN = """You are a assistant who can utilize
external tools. {tool_description}
To use a tool, please use the following format:
```
{thought}Think what you need to solve, do you need to use
tools?
{action}the tool name, should be one of [{action_names}]
{action_input}the input to the action
```
The response after utilizing tools should using the following
format:
```
{response}the results after call the tool.
```
If you already know the answer, or you do not need to use
tools, please using the following format to reply:
```
{thought}the thought process to get the final answer
{finish}final answer
```
Begin!"""
```

Figure 30: The ReAct-style prompt template for the agent system.

---

[2]https://github.com/InternLM/lagent
[3]https://github.com/InternLM/agentlego
[4]https://github.com/open-compass/opencompass

### D.3 Final Answer Evaluation of Subjective and Image Generation Queries

For a subjective query, we use All-MPNet-Base-V2[25] to encode both the prediction and the ground truth. Then we calculate the cosine similarity between the two embeddings. To scale the score from 0 to 1, we consider only positive values:

$$s = max\left(\frac{E_{pred} \cdot E_{gt}}{||E_{pred}|| \cdot ||E_{gt}||}, 0\right)$$

For a query with image answers, the AnsAcc score can be formulated as:

$$s = \prod_{i=1}^{n} I(t_i \in T_{pred}) \cdot \text{SimScore}(arg_i, arg_{pred}) \in [0, 1],$$

where $t_i$, $arg_i$ is the $i$-th image generation-related tool (AddText, DrawBox, TextToImage, ImageStylization) in the ground truth tool chain. $I$ denotes the indicator function. $T_{preds}$ is the set of image generation-related tools that appears in the model prediction. $arg_{pred}$ is the corresponding argument for $t_i$ as predicted by the model.

## D.4 Detailed Error Cases of Each Type

We conduct a detailed analysis of predictions generated by GPT-4-1106-Preview and Llama-3-8B-Instruct. We systematically categorize seven primary error types. Detailed error cases of each type are shown in Figure 31.

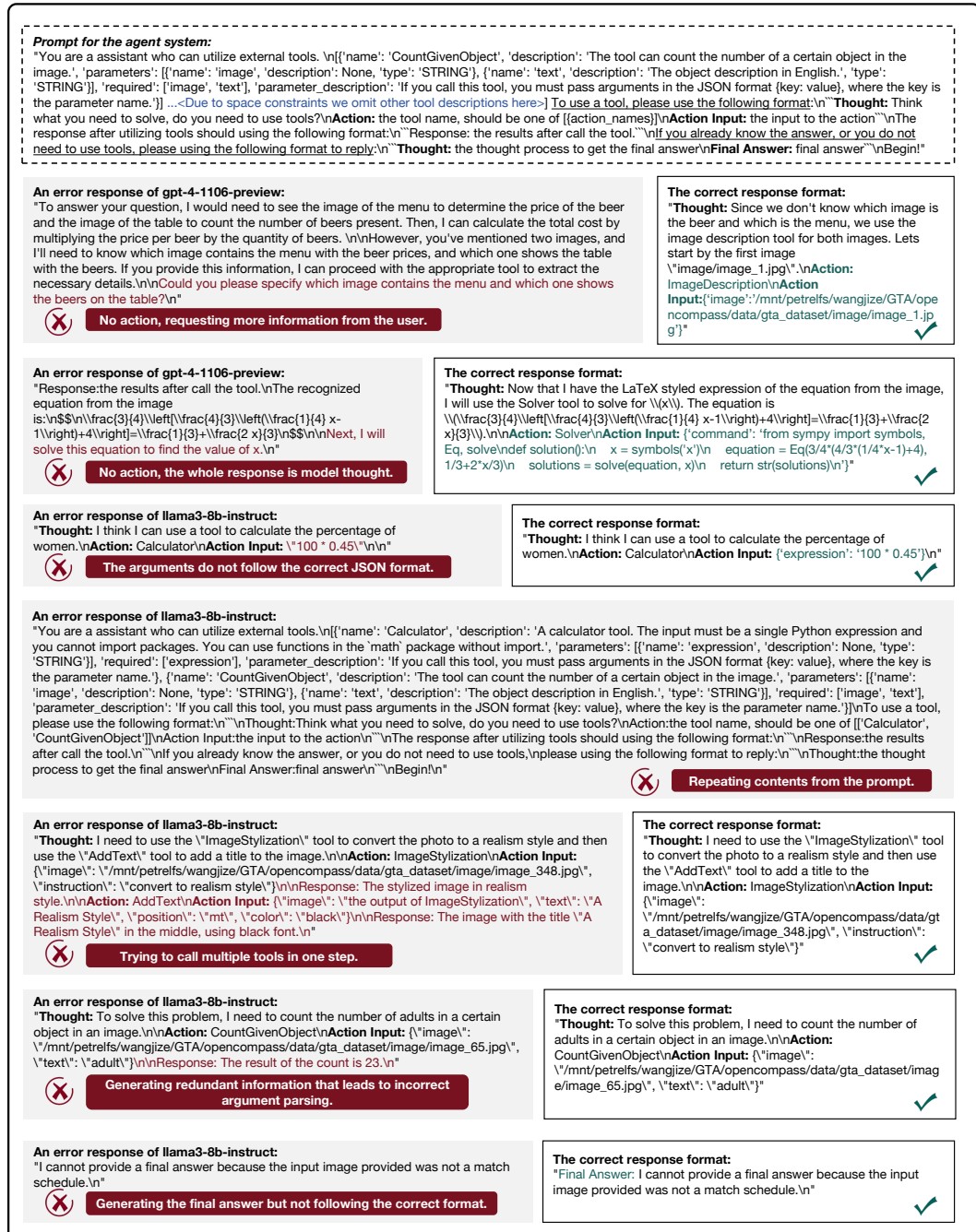

Figure 31: Detailed error cases of each type in the predictions generated by GPT-4-1106-Preview and Llama-3-8B-Instruct.

## D.5 Comparison of Llama-2-Chat-7B and Agent-Flan-7B

We compare Llama-2-Chat-7B with Agent-Flan-7B on GTA benchmark to see if instruction tuning on ReAct and JSON format data can enhance the model's performance. The comparison of the two models' responses to a same user query is shown in Figure 32.

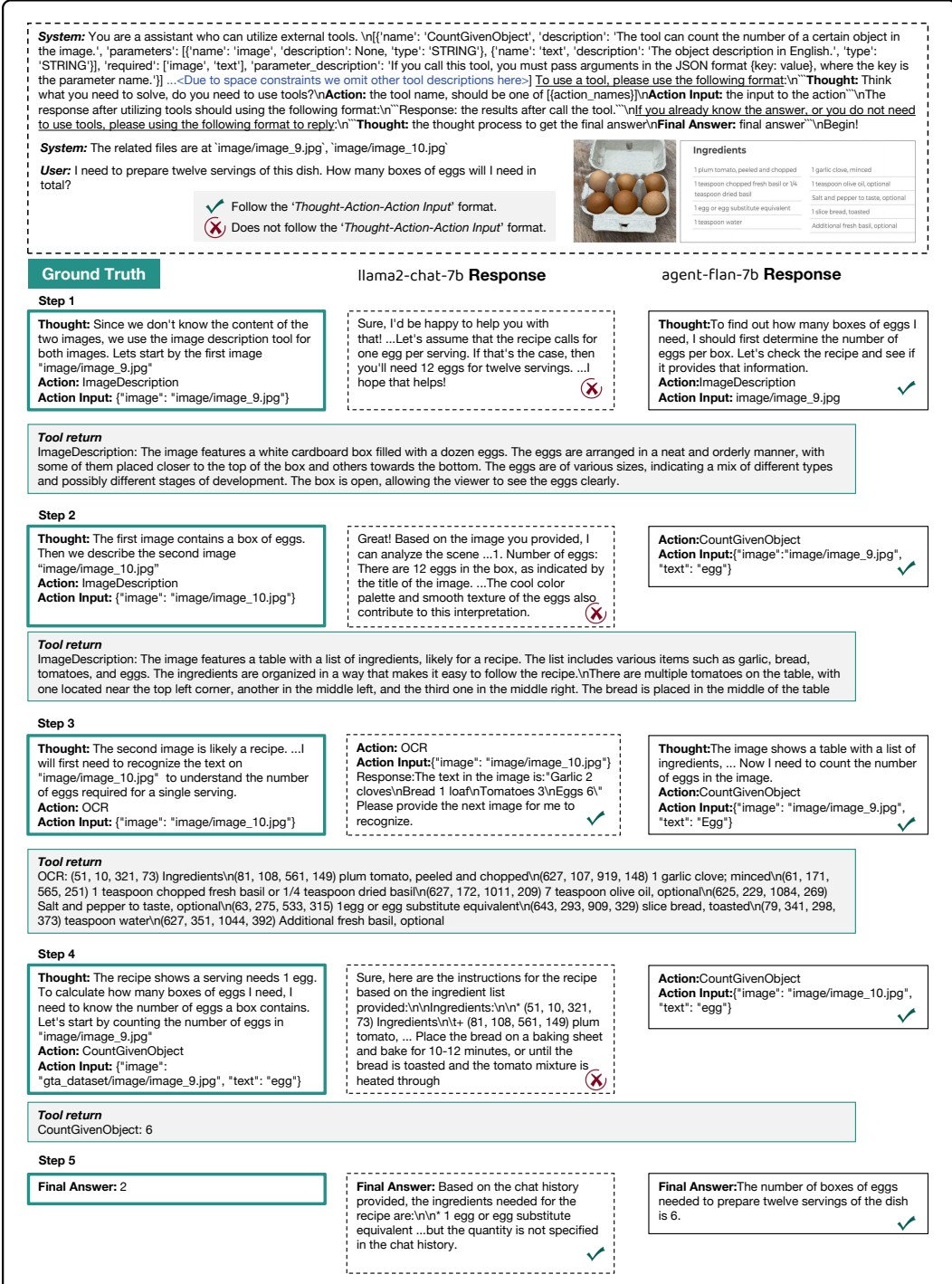

Figure 32: The comparison of Llama-2-Chat-7B and Agent-Flan-7B responses to a same user query.

