# OpenReview forum: "GTA: A Benchmark for General Tool Agents"
_NeurIPS.cc/2024/Datasets_and_Benchmarks_Track — NeurIPS 2024 Track Datasets and Benchmarks Poster_

### Official Review · Reviewer_kUMt · 2024-07-23
**Review of Paper 1037**

**Rating:** 7
**Confidence:** 4
**Correctness:** Yes
**Clarity:** Yes

**Review:**

Tool use is important. In this paper, the authors try to benchmark the capabilities of tool use in real-world scenarios. The paper is well-motivated, clear written and novel.

**Strengths:**

1. Real-world relevance: The benchmark uses human-designed queries based on real-world scenarios, which is more realistic.

2. Comprehensive tool set: GTA incorporates 14 tools across perception, operation, logic, and creativity categories, allowing for a more holistic evaluation of LLM capabilities.
3. Multimodal inputs: By including image files as context for queries, the benchmark better aligns with real-world problem-solving scenarios.
4. Fine-grained evaluation: The authors propose multiple metrics to assess different aspects of tool use, from instruction following to argument prediction and final answer accuracy.
5. Extensive evaluation: The paper presents results from testing 16 different LLMs, providing valuable insights into the current state of tool-use capabilities across various models.
6. Reproducibility: The authors have made the dataset and evaluation platform publicly available, which is commendable for fostering further research in this area.

**Additional Feedback:**

1. The current benchmark is still static, however, the tool use scenarios may be dynamic. Could the authors include more discussion about whether the current benchmark can be extended to include dynamics scenarios?
2. The human-written nature of the queries and tool chains, while ensuring high quality, poses challenges to scalability. As AI capabilities evolve rapidly, there may be a need to frequently update or expand the benchmark. The current approach might make such updates time-consuming and expensive. How to handle this?

**Documentation:**

Yes

**Limitations:**

Yes

**Opportunities For Improvement:**

Please see the additional feedback

**Relation To Prior Work:**

Yes

**Summary And Contributions:**

This paper introduces GTA (General Tool Agents), a novel benchmark designed to evaluate the tool-use capabilities of large language models (LLMs) in real-world scenarios. The authors strongly motivate the need for such benchmark, highlighting the limitations of existing evaluations that often rely on AI-generated queries, single-step tasks, and text-only inputs.

---

> ### Author Rebuttal · Authors · 2024-08-17
>
> **Q1: Whether the current benchmark can be extended to include dynamics scenarios?**
>
> We appreciate the reviewer's thoughtful feedback. In fact, the end-to-end mode of GTA evaluation supports a dynamic evaluation of tool use, where the model plans and calls tools freely and dynamically without the constraint of predetermined steps. In this mode, only the final answer is evaluated. Besides, dynamic tool use scenarios might involve an expanded set of tools. In this case, API retrieval methods could also be integrated to retrieve suitable tools. We will include the discussion about tool use evaluation in dynamic scenarios in our revised paper.
>
> **Q2: The current hand-written approach makes frequent updates or expansions expensive and time-consuming.**
>
> We thank the reviewer for the valuable feedback. Utilizing AI-generated data presents a potential strategy for cost reduction. Nevertheless, due to the limited capabilities of current LLMs, even for the most advanced models such as GPT-4o, there remains considerable challenges to generate data for complex real-world tasks in GTA.  Data of substandard quality may pose a risk to a faithful evaluation. If we consider integrating AI generated data, it is imperative to develop more robust data construction algorithms to safeguard the benchmark's credibility. Besides, the community of complex task evaluation requires more capable LLMs to provide a foundation for reliable data generation.

---

### Official Review · Reviewer_Guc9 · 2024-07-25
**Review Comment**

**Rating:** 5
**Confidence:** 3
**Correctness:** Mostly Correct.
**Clarity:** Mostly well written.

**Review:**

My main concern is about the scope being (potentially) too big to cover. The authors claim to cover "general" to agents and real-world problem solving, to achieve which the 229 constructed real-world tasks are still not enough in their scope. Also the number of queries and images might not capture as many as real-world scenarios. Suggesting the authors consider narrow the claims and focus on presenting the matters they care the most.S

1.	Please provide a more detailed description of the evaluation platform, including the testing process for the LLMs and the prompts they receive.
2.	GTA comprises 229 questions, which can be categorized into various classes based on different capabilities. The number of questions per category may be too small to accurately reflect the true capabilities of LLMs.
3.	The paper introduces new evaluation standards and reveals the existing bottlenecks of LLMs but does not present concrete results from improved models. In other words, the benchmark should provide a more correct answer to inform other researchers how to modify LLMs to achieve better results. It would be beneficial to include results from an enhanced LLM that performs better under the GTA criteria compared to its original model.

**Strengths:**

The use of real user queries and real deployed tools provides an assessment of LLM capabilities in practical scenarios.

Incorporating authentic image files for query contexts enhances the relevance and difficulty of the tasks.

The paper introduces fine-grained metrics (InstAcc, ToolAcc, ArgAcc, SummAcc, and AnsAcc) for assessing various aspects of tool-use capabilities.

Covers tools across perception, operation, logic, and creativity categories

**Additional Feedback:**

N.A.

**Documentation:**

Yes.

**Ethics:**

N.A.

**Limitations:**

N.A.

**Opportunities For Improvement:**

Please refer to the review

**Relation To Prior Work:**

Yes.

**Summary And Contributions:**

The paper introduces a benchmark called GTA designed to evaluate the tool-use capabilities of large language models (LLMs) in real-world scenarios. The benchmark features three key components: human-written queries, real deployed tools, and multimodal inputs. It emphasizes the importance of reasoning and planning in using tools to solve real-world tasks. The study evaluates 16 mainstream LLMs on 229 real-world tasks, revealing gaps in the current tool-use capabilities of LLMs, with even advanced models like GPT-4 completing fewer than 50% of the tasks. The findings highlight the need for improvements in argument prediction and other tool-use aspects to advance the development of general-purpose AI agents.

---

> ### Author Rebuttal · Authors · 2024-08-17
>
> **Q1: The scope of 'general' and 'real-world' is too big to cover with only 229 tasks.**
>
> We appreciate the thoughtful feedback from the reviewer. We discuss this concern in the general response '*Q1: Data size and diversity*'.
>
> **Q2: Provide a more detailed description of the evaluation platform, including test process and prompts.**
>
> We are grateful for the reviewer's suggestion. We use Lagent as the agent framework. Experiments are conducted on the Opencompass evaluation platform. During the test process, the LLM serves as a central controller:
> 1. First, the LLM receives the system prompt, including the agent protocol that specifies chatting format (we use the ReAct protocol for the test process), and a tool list that contains tool names, functions and parameter formats.
> 2. Then, the LLM receives the user query and paths of the input images.
> 3. After that, the LLM is expected to generate responses that aim at either calling tools or giving a final answer.
> 4. The agent system will parse the tool names, arguments, and final answers from the LLM responses according to the agent protocol in the system prompt.
> 5. Then the agent system executes tools, sending execution results or execution errors to the LLM.
> 6. The LLM will decide the next step based on the dialog history and tool responses.
>
> The system prompt for the LLMs contains the tool list, the agent protocol (chatting format), and the related file paths:
>
> ```
> [{'role': 'system', 'content': "You are a assistant who can utilize external tools.\n [{'name': 'OCR', 'description': 'This tool can recognize all text on the input image.', 'parameters': [{'name': 'image', 'description': None, 'type': 'STRING'}], 'required': ['image'], 'parameter_description': 'If you call this tool, you must pass arguments in the JSON format {key: value}, where the key is the parameter name.'}, {'name': 'CountGivenObject', 'description': 'The tool can count the number of a certain object in the image.', 'parameters': [{'name': 'image', 'description': None, 'type': 'STRING'}, {'name': 'text', 'description': 'The object description in English.', 'type': 'STRING'}], 'required': ['image', 'text'], 'parameter_description': 'If you call this tool, you must pass arguments in the JSON format {key: value}, where the key is the parameter name.'}, ...<Due to space constraints we omit other tool descriptions here>]
> To use a tool, please use the following format:
> ```Thought:Think what you need to solve, do you need to use tools?
> Action:the tool name, should be one of ['OCR', 'CountGivenObject', ...]
> Action Input:the input to the action```
> The response after utilizing tools should using the following format:
> ```Response:the results after call the tool.```
> If you already know the answer, or you do not need to use tools, please using the following format to reply:
> ```Thought:the thought process to get the final answer\n Final Answer:final answer```
> Begin!"},
> {'role': 'system', 'content': 'The related files are at `gta_dataset/image/image_1.jpg`, `gta_dataset/image/image_2.jpg`'},
> {'role': 'user', 'content': 'How much should I pay for the beer on the table according to the price on the menu?'}]
> ```
>
> To evaluate LLMs on GTA dataset, one should first prepare the dataset, launch a model service, and deploy all tools. More detailed test process information could be found in the GTA Github page.
>
> **Q3: The number of questions per category may be too small.**
>
> We appreciate the reviewer's thoughtful feedback. In our experiments, we do not divide the **questions** in different categories, but divide **tool calls** in different categories and evaluate their F1 scores. Since our tasks are multi-step, the total step number reaches 728, thus the tool calls for each category are of a reasonable amount.
>
> **Q4: The benchmark should provide a more correct answer to inform other researchers how to modify LLMs to achieve better results. It would be beneficial to include results from an enhanced LLM that performs better under the GTA criteria compared to its original model.**
>
> We thank the reviewer for the valuable feedback. We provide the detailed error analysis and further exploration to improve LLMs in the general response '*Q2: Detailed error analysis*'.

---

### Official Review · Reviewer_hvCE · 2024-07-25

**Rating:** 7
**Confidence:** 3
**Correctness:** Yes, the claims in this paper are cor…
**Clarity:** Yes, it's well written.

**Review:**

Quality: 8/10
* Pro: The study was well-designed, with clear motivation and solid implementation. Its codebase is also well-structured.

Clarity: 9/10
* Pro: The paper is well-written and easy to follow.

Originality: 8/10
* Pro: The contributions of this paper are original.

Significance: 8/10
* Pro: This study provides a human-annotated dataset and an evaluation platform with well-written codes. Popular LLMs have been evaluated in this paper. Combining these, I believe it would make a considerable impact on the research community.

**Strengths:**

* **A human-annotated dataset**: The dataset in this study was annotated by humans, making it closer to real-world scenarios and confirming the quality of the dataset.

* **An evaluation platform**: The designed evaluation ways and evaluation platform make it easy for the follow-up works to use this benchmark.

* **The task is still challenging**: The benchmarking results showed that it's still a challenging benchmark for existing LLM-based agents, revealing the drawbacks of existing agents.

**Additional Feedback:**

N/A

**Documentation:**

Yes, the data detail is sufficient.

**Ethics:**

No.

**Limitations:**

Yes, the manuscript clearly addresses these.

**Opportunities For Improvement:**

* **Lacking the metrics for image generation**: To make the evaluation complete, a solution can be proposed to evaluate the image generation tasks for AnsAcc.

* **Relations to existing studies**: Although they might be concurrent works like [1], it would be good to discuss the relations to them.

* **Minor**: In Line 229, "cosine similarity" => Describe the process of computing cosine similarity, including the embedding method.

[1] Xie, Tianbao, et al. "Osworld: Benchmarking multimodal agents for open-ended tasks in real computer environments." arXiv preprint arXiv:2404.07972 (2024).

**Relation To Prior Work:**

Largely yes.

**Summary And Contributions:**

**Topic**: LLM-based agents

**Input**: files (one or two images), a toolset, and the query

**Output**: Toolchain and final answer

**Summary and contribution**:
* Dataset: The paper introduced a dataset for LLM-based agents containing 229 real-world tasks. Compared with existing benchmarks, the proposed one is closer to real-world scenarios, including user-designed queries, real deployed tools, and human-annotated toolchains.
* Evaluation: The paper proposed two ways to evaluate agents, including end-to-end mode and step-by-step mode, thanks to the annotation.
* Benchmark: It benchmarked a set of LLMs on this task, showing that it's a challenging task for existing agents.

---

> ### Author Rebuttal · Authors · 2024-08-17
>
> **Q1: Evaluate the image generation tasks for AnsAcc.**
>
> We thank the reviewer for the valuable suggestions. There are primarily two potential methods to evaluate the AnsAcc of image generation tasks. The first method involves utilizing a multimodal model like GPT-4o to serve as a judge model, which is instructed to assign scores to generated images. However, the impartiality and evaluative capability of such a model cannot be guaranteed, casting doubts on the reliability of this evaluation technique.
>
> The alternative method entails an indirect evaluation of the generated images, by analyzing the tools and their arguments. Given that the outcome of the image generation is determined solely by the input parameters, evaluating the accuracy of these parameter predictions can effectively reflect the appropriateness of the resultant images. If the predicted parameters are correct, the images produced should align with the specified task objectives. Consequently, due to its more direct and plausible assessment, we opt for the second method to evaluate AnsAcc for image generation tasks.
>
> For an image generation task, the score can be formulated as:
>
> $$ s=\prod\limits_{i=1}^{n} I(t_i \in T_{pred})\cdot \text{SimScore}(arg_i, arg_{pred}) \in [0, 1] $$
>
> where $t_i, arg_i$ is the $i$-th image generation-related tool (AddText, DrawBox, TextToImage, ImageStylization) in the ground truth tool chain. $I$ denotes the indicator function. $T_{pred}$ is the set of image generation-related tools that appears in the model prediction. $arg_{pred}$ is the corresponding argument for $t_i$ as predicted by the model.
>
> We revise the AnsAcc results in the subsequent table. After considering the scores from image generation tasks, we observe a general decline in the AnsAcc of most models, while that of GPT-4o experience an increase. This indicates that GPT-4o is notably more adept at choosing tools and arguments for image generation tasks, in contrast to the majority of models that exhibit shortcomings in this area.
>
> | Model                    | AnsAcc (w/o ImgGen) | AnsAcc (w/ ImgGen) |
> |--------------------------|---------------------|--------------------|
> | gpt-4-1106-preview               | 46.59               | 44.9               |
> | gpt-4o                   | 38.73               | 40.05              |
> | gpt-3.5-turbo            | 21.47               | 21.18              |
> | claude-3-opus            | 18.97               | 14.47              |
> | mistral-large     | 14.82               | 11.94              |
> | qwen1.5-72b-chat         | 13.32               | 10.22              |
> | qwen1.5-14b-chat              | 12.42               | 9.33               |
> | qwen1.5-7b-chat          | 10.56               | 7.93               |
> | mixtral-8x7b-instruct    | 9.77                | 7.34               |
> | deepseek-llm-67b-chat | 9.51 | 7.93 |
> | llama3-70b-instruct   | 8.32 | 6.25 |
> | mistral-7b-instruct   | 7.37 | 5.54 |
> | deepseek-llm-7b-chat      | 4    | 3.01 |
> | yi-34b-chat                | 3.21 | 2.41 |
> | llama3-8b-instruct    | 3.1  | 2.74 |
> | yi-6b-chat                 | 0.58 | 0.44 |
>
>
> **Q2: Discuss the relations with works like Osworld.**
>
> We appreciate the reviewer's recommendation and will incorporate a discussion on the relationship between GTA and benchmarks such as Osworld in our revised paper. Osworld serves as a benchmark to evaluate the capabilities of autonomous agents in completing intricate computer tasks, which require interaction with actual web and desktop applications. GTA is a benchmark to evaluate general tool-augmented agents on close-to-real-world tasks. Both GTA and Osworld feature multi-step, complex tasks inspired by authentic user cases, and both engage with multimodal environments and executable platforms. However, Osworld is specifically tailored for open-ended tasks in real computer environments, whereas GTA is devised for tool agents operating in more generalized real-world scenarios.
>
> **Q3: Describe the process of computing cosine similarity, including the embedding method.**
>
> We are grateful for the reviewer's suggestion and will detail the computation of cosine similarity in the revised manuscript. Using All-MPNet-Base-V2[1], we encode both the prediction and the ground truth, followed by the calculation of the cosine similarity between the two embeddings. To scale the score from 0 to 1, we consider only positive values:
>
> $$ s=max(\frac{E_{pred} \cdot E_{gt}}{||E_{pred}|| \cdot ||E_{gt}||}, 0) $$
>
> [1] Song, Kaitao, et al. "Mpnet: Masked and permuted pre-training for language understanding." *Advances in neural information processing systems* 33 (2020): 16857-16867.

---

### Official Review · Reviewer_vpCb · 2024-07-26
**The paper presents a significant advancement in evaluating the real-world tool-use capabilities of LLMs**

**Rating:** 6
**Confidence:** 3

**Review:**

The paper offers a comprehensive benchmark, GTA, which represents a significant advancement in evaluating the real-world tool-use capabilities of LLMs. The benchmark’s design, which includes real user queries, deployed tools, and multimodal inputs, addresses critical gaps in current evaluations that often rely on AI-generated queries and virtual tools. This setup more accurately reflects real-world scenarios, providing valuable insights into the limitations and capabilities of existing LLMs. The clarity of the paper is commendable, with well-structured sections and detailed descriptions of the methods and results. The paper’s findings highlight the need for developing more robust reasoning and planning capabilities in LLMs, especially for tasks that involve tool use. However, the number and diversity of tasks and inputs in the dataset may not be sufficient to cover all possible real-world scenarios. With a limited dataset, the evaluation results might be biased towards certain types of tasks, failing to provide a comprehensive assessment of model performance. Additionally, while the paper provides a broad evaluation of various LLMs, further exploration of the reasons behind specific failures or successes would enrich the analysis.

**Strengths:**

- The use of real-world user queries and deployed tools makes GTA a realistic and challenging benchmark for evaluating LLMs.
- The paper is well-written and logically organized. The methodology and results are presented clearly.
- The inclusion of multimodal inputs provides a comprehensive evaluation of LLMs’ capabilities beyond text-only tasks.
- The detailed evaluation metrics and analysis offer deep insights into the current limitations of LLMs in tool-use tasks.

**Additional Feedback:**

The GTA benchmark is a valuable contribution to the field, offering a realistic and challenging evaluation of LLM capabilities. The paper could be strengthened by including a discussion on the broader implications of the findings, particularly concerning the development of more advanced LLMs and their potential societal impact. Additionally, providing a more detailed analysis of model failures would help in understanding the limitations of current LLMs and guide future research efforts. Moreover, exploring additional domains and expanding the diversity of tasks could further enhance the benchmark’s value.

**Clarity:**

The paper is well-written, with clear and concise explanations of the benchmark’s design, methodology, and findings. The structure is logical, and the flow of information is easy to follow.

**Correctness:**

The claims made in the paper appear to be correct, based on the presented data and evaluation. The benchmark is constructed soundly, and the experiments are performed correctly, providing a robust assessment of the LLMs’ tool-use capabilities.

**Documentation:**

The paper provides sufficient detail on the construction and organization of the dataset, including the types of tasks, tools, and evaluation metrics used. However, the documentation could be improved by including more information on the ethical guidelines followed during the data collection process.

**Ethics:**

The paper does not explicitly address ethical concerns related to the use of the benchmark, such as data privacy or the potential misuse of tool-augmented LLMs.

**Limitations:**

The human-designed nature of the dataset and tool chains, while ensuring high quality, may not fully capture the variety of real-world scenarios. The authors do not discuss the potential negative societal impacts of their work, which is crucial for understanding the broader implications of deploying such systems in real-world applications.

**Opportunities For Improvement:**

- The dataset and tool chains are human-designed, which may limit scalability. Incorporating more AI-generated content while maintaining quality could enhance the benchmark’s scope.
 - The paper provides a general overview of the models’ performance but lacks a detailed error analysis. Understanding why certain models fail on specific tasks could offer valuable insights into their limitations and areas for improvement.

**Relation To Prior Work:**

The paper effectively situates its contribution within the context of existing benchmarks and tool-use evaluations. It clearly delineates how GTA differs from and improves upon previous benchmarks, particularly in its focus on real-world scenarios and comprehensive tool-use evaluation.

**Summary And Contributions:**

The paper introduces GTA, a benchmark designed to evaluate the tool-use capabilities of LLMs in real-world scenarios. The benchmark addresses gaps in current tool-use evaluations by incorporating real-world user queries, real deployed tools, and multimodal inputs. The authors design 229 tasks that require LLMs to use various tools in categories like perception, operation, logic, and creativity. The evaluation reveals significant challenges for current LLMs, with even advanced models like GPT-4 achieving less than 50% task completion.

---

> ### Author Rebuttal · Authors · 2024-08-17
>
> **Q1: The number and diversity may not be sufficient to cover all possible real-world scenarios.**
>
> We appreciate the thoughtful feedback from the reviewer. We discuss this concern in the general response '*Q1: Data size and diversity*'.
>
> Besides, we thank for the insightful suggestion that incorporating more AI-generated content while maintaining quality could enhance the benchmark’s scope. We agree that it is indeed a potential data construction approach. However, due to the limited capabilities of current LLMs, even for the most advanced models such as GPT-4o, there remains considerable challenges to generate data for complex real-world tasks in GTA. Data of substandard quality may pose a risk to a faithful evaluation. The community of complex task evaluation requires more robust models to provide a foundation for reliable data generation.
>
> **Q2: Further exploration of the reasons behind specific failures or successes would enrich the analysis.**
>
> We appreciate the reviewer's valuable suggestions. We provide the detailed error analysis in the general response '*Q2: Detailed error analysis*'.
>
> **Q3: The authors do not discuss the potential negative societal impacts of their work.**
>
> We express our gratitude for the reviewer's suggestions. The GTA benchmark may have potential negative societal impacts. These include copyright concerns related to image data collection, as detailed in Supplementary Material A.2. The presence of images involving people in our dataset also raises privacy concerns. Additionally, during the evaluation of GTA, the agent system could potentially experience hallucinations and generate harmful information. Besides, given the inclusion of coding questions in GTA, the agent system might produce malicious code. We will elaborate on these potential negative societal impacts in our revised paper.

---

### Author Rebuttal · Authors · 2024-08-17

# General Response

We sincerely appreciate the reviewers' great efforts in reviewing this paper. We thank all reviewers for their helpful feedback.

We are glad that the reviewers found our benchmark "realistic and challenging" [vpCb,hvCE,Guc9,kUMt], "comprehensive and robust" [vpCb,kUMt], offering "fine-grained" evaluation metrics [vpCb, Guc9, kUMt], our paper "well-written and logically organized" [vpCb, hvCE, Guc9, kUMt], the code implementation "well-structured and solid" [hvCE], "commendable for fostering further research in this area" [kUMt], and "would make a considerable impact on the research community" [hvCE].

The main concerns we attempt to address are about the data size and diversity, and the detailed error analysis, as provided below.

## Q1: Data size and diversity.

We are grateful for the reviewers' insightful comments and constructive suggestions. There may be concerns regarding the data size and diversity. We wish to highlight that our dataset exhibits remarkable diversity and has a reasonable scale to fulfill the requirements of the GTA evaluation.

The GTA dataset demonstrates significant diversity across several dimensions:
  - **Scenario Diversity**: It encompasses a broad spectrum of scenarios reflective of real-world challenges, such as multi-image reasoning, diagram analysis, coding, visual interaction, web browsing, mathematics, and creative arts. This breadth supports the comprehensive assessment of real-world problem-solving abilities.
  - **Tool Diversity**: The dataset includes tools categorized into perception, operation, logic, and creativity, representing four indispensable capabilities of a tool agent to solve real-world tasks. The tool chains are combinations of multiple tools from different categories, providing a vast and flexible solution space.
  - **Input Diversity**: A range of authentic multimodal inputs, including spatial scenes, web screenshots, tabular data, code snippets, and printed/handwritten materials, are present in the dataset. Additionally, user queries vary in linguistic expression, closely mirroring real-world applications.

Besides, the data scale of GTA dataset is reasonable:

Evaluating agent systems in real-world scenarios poses great challenges in dataset collection, including collecting non-trivial and practical tasks, annotating fine-grained tool-invoking processes and building automatic evaluation systems with all tools deployed. Hence, existing work within the community places greater emphasis on data quality and evaluation methodologies than on data volume, as illustrated in the table below.

| Benchmark    | Purpose                                                                                                                      | Construction Method | Size |
|--------------|------------------------------------------------------------------------------------------------------------------------------|------|------|
| API-Bank [1] | Evaluate LLMs' capabilities in planning, retrieving, and calling APIs.                                                       | Manually annotated | 314  |
| GAIA [2]     | Evaluate general AI assistants with questions that are conceptually simple yet challenging for AIs. | Manually annotated | 466  |
| ToolEmu [3]  | Identify the risks of LM agents with an LM-emulated sandbox.                                                                 | GPT-4 generated + manually verified | 144  |
| OSWorld [4]  | Evaluate agents for open-ended tasks in real computer environments.                                                          | Manually annotated | 369  |
| GTA (Ours)   | Evaluate tool-augmented agents on more generalized and close-to-real-world tasks.                                            | Manually annotated | 229  |

  [1] Li, Minghao, et al. "API-Bank: A Comprehensive Benchmark for Tool-Augmented LLMs." *Proceedings of the 2023 Conference on Empirical Methods in Natural Language Processing.* 2023.

  [2] Mialon, Grégoire, et al. "GAIA: a benchmark for General AI Assistants." *The Twelfth International Conference on Learning Representations.* 2024.

  [3] Ruan, Yangjun, et al. "Identifying the Risks of LM Agents with an LM-Emulated Sandbox." *The Twelfth International Conference on Learning Representations.* 2024.

  [4] Xie, Tianbao, et al. "Osworld: Benchmarking multimodal agents for open-ended tasks in real computer
  environments." *arXiv preprint arXiv:2404.07972.* 2024.

These studies share two common features: (1) **high task complexity**, and (2) **high data quality**. As for GTA, on one hand, the task is complex, involving multiple tools, multiple steps, and multimodality. On the other hand, the data quality is high. The annotation method of GTA ensures that the questions are close to real-world scenarios, the tool chains are executable, and the final answers are unique and accurate. Moreover, diversified expansion strategies are adopted to maximize the differences in scenarios between questions. These requirements ensure the data quality of GTA.

The scale of our GTA dataset is comparable to these studies within the community and is widely agreed to be sufficient to reflect and compare agents' capabilities. Consistency in certain trends observed in our experiments, such as larger models from the same series outperforming their smaller counterparts, further confirms that our data can support reliable evaluations.

---

> ### Author Rebuttal · Authors · 2024-08-17
>
> ## Q2: Detailed error analysis.
>
> We acknowledge the reviewer's valuable suggestions. In Section 4.3, we discuss the bottleneck in task performance arising from most models' inability to generate responses or predict arguments in the correct format. To understand the reason behind these model failures, we conduct a detailed analysis of the predictions generated by ```gpt-4-1106-preview``` and ```llama3-8b-instruct```.
>
> We systematically categorize seven primary error types. The statistical outcomes are presented in the following table. Detailed error cases of each type can be found **in the accompanying PDF file**.
>
> | Index | Error Type                                                                 | gpt-4-1106-preview | llama3-8b-instruct   |
> |-------|----------------------------------------------------------------------------|-------------------|-------------|
> |       | **Planning error:**                                                            |                   |             |
> | 1     | No action, requesting more information from the user.                      | 24 (38.7%)        | 0           |
> | 2     | No action, the whole response is model thought.                            | 31 (50%)          | 2 (0.1%)    |
> |       | **Format error:**                                                              |                   |             |
> | 3     | The arguments do not follow the correct JSON format.                       | 5 (8.1%)          | 118 (45.4%) |
> | 4     | Trying to call multiple tools in one step.                                 | 1 (1.6%)          | 43 (16.5%)  |
> | 5     | Generating redundant information that leads to incorrect argument parsing. | 0                 | 51 (19.6%)  |
> | 6     | Repeating contents from the prompt.                                        | 0                 | 41 (15.8%)  |
> | 7     | Generating the final answer but not following the correct format.          | 1 (1.6%)          | 5 (1.9%)    |
> |       | **Total Number**                                                               | 62 (100%)         | 260 (100%)  |
>
> Our analysis reveals distinct error distributions between GPT-4 and LLaMA-3. GPT-4 consistently adheres to the given prompts when executing actions, in contrast to LLaMA-3, which often fails to maintain the prescribed format. However, GPT-4 is prone to generating passive thought processes or attempting interaction with the user, rather than taking decisive action.
>
> For the GPT-4 model, the predominant type of error is No Action, wherein the model neither utilizes tools nor produces a final answer. In 38.7% of erroneous responses, GPT-4 attempts to engage with the user, mistakenly assuming the query lacks clarity and requesting additional information, despite the query and input images supplying sufficient details for task resolution. Furthermore, 50% of the error responses consist solely of the model's internal thought without any corresponding action.
>
> For the LLaMA-3 model, the majority of errors are related to formatting during action sequences, such as invoking tools or generating the final answer. Specifically, 45.4% of errors originate from argument predictions not adhering to a valid JSON format. Additionally, in 16.5% of the flawed responses, the model attempts to invoke multiple tools simultaneously, a function not supported by the agent system. Moreover, 19.6% of the errors occur when the model disregards the prompt and generates redundant information after argument prediction, leading to incorrect argument parsing. Finally, in 15.8% of the cases, the model fails to perform the correct action, merely repeating content from the prompt.
>
> To enhance model performance under GTA criteria, we propose several promising directions for exploration:
>
>   - **Instruction-tuning on ReAct and JSON format data.** Since the LLM functions as the central controller of an agent system, producing responses that strictly comply with the agent protocol is important. Fine-tuning on ReAct and JSON format may mitigate format-related errors during action execution.
>
>   - **Efficient in-context learning methods.** Given that the test prompt is presented in a zero-shot manner, the LLM may have difficulties in fully understanding the format instructions. In-context learning may address this challenge by giving LLMs additional examples. Due to the complexity of the task, however, the example tends to be long and flexible. How to design efficient in-context learning methods needs further investigation.
>
>   - **Self-reflection mechanisms for error correction.** Self-reflection mechanisms could be applied to enable models correct their errors according to the feedback from tools or external environments. Thus the model can reanalyze the query and make new plannings or actions, which may solve the No Action errors occurring in GPT-4's predictions.

---

> > ### Author Rebuttal · Authors · 2024-08-23
> >
> > Among these aspects, we further conduct preliminary verification to see if **instruction tuning on ReAct and JSON format data** can enhance the model's performance on the GTA dataset. AgentFLAN [1] is a popular instruction tuning method that fine-tunes LLM-based agents using ReAct and JSON instruction-following data. We compare LLaMA2-chat-7B with Agent-Flan-7B (which is fine-tuned from LLaMA2-chat-7B) on GTA benchmark.
> >
> > | Model | InstAcc        | ToolAcc | ArgAcc | SummAcc |
> > |----------------|---------|--------|---------|------|
> > | llama2-chat-7b | 30.59   | 13.82  | 0.54    | **49.14** |
> > | agent-flan-7b  | **68.59**   | **39.86**  | **6.64**    | 48.47 |
> >
> > Under the step-by-step mode, we discovered that Agent-Flan-7B's InstAcc and ToolAcc metrics were significantly higher than those of LLaMA2-chat-7B. We provide the comparison of the two models' responses to a same user query **in the accompanying PDF file**. The responses of Agent-Flan-7B follow the format of '*Thought-Action-Action Input*' that specified in the prompt. But most responses of LLaMA2-chat-7B fail to follow the format. This further suggests the improved instruction following capability of Agent-Flan-7B.
> >
> > However, we note that the improvement in ArgAcc is not significant. In the response comparison in the PDF file, we find that although Agent-Flan-7B follows the format of '*Thought-Action-Action Input*', it sometimes fails to generate the argument (Action Input) in a correct JSON format, or summarizes the final answer incorrectly. Thus, how to further enhance the model's capabilities on GTA through instruction fine-tuning is still an open problem. We will discuss potential methods to improve the model's performance on GTA in the revised version of the paper.
> >
> > [1] Chen, Zehui, et al. "Agent-FLAN: Designing Data and Methods of Effective Agent Tuning for Large Language Models." *Findings of the Association for Computational Linguistics ACL* (2024).

---

### Author Response · Authors · 2024-08-25

Dear Reviewers and AC,

Your constructive advice and valuable comments really help improve our paper. We have tried our best to address the mentioned concerns/problems in the rebuttal. Feel free to let us know if there is anything unclear or so. We are happy to clarify them.

Best, Authors

---

### Author Response · Authors · 2024-08-30

Dear Reviewers and AC,

We sincerely appreciate your great efforts in reviewing this paper. Your constructive advice and valuable comments really help improve our paper. Considering the approaching deadline, please, let us know if you have follow-up concerns. We sincerely hope you can consider our reply in your assessment, and we can further address unclear explanations and remaining concerns if any.

Once more, we are appreciated for the time and effort you've dedicated to our paper.

Best, Authors

---

### Author Response · Authors · 2024-09-01
**Looking Forward to Your Kind Response**

Dear Reviewers and AC,

We sincerely appreciate your thorough review and constructive feedback on our paper. As the deadline approaches, please, let us know if you have follow-up concerns. We hope our responses will be taken into account in your final assessment, and we are ready to provide further clarification on any remaining issues if needed. Once again, we are grateful for the time and effort you have invested in evaluating our paper. Looking forward to your kind response.

Best regards,
The Authors

---

### Decision · Program_Chairs · 2024-09-26

**Decision:**

Accept (Poster)

**Comment:**

After the rebuttal period, the paper received two acceptances, one borderline acceptance, and one borderline rejection. While R3 expressed concerns about the dataset's scope, arguing that 229 real-world tasks may not be sufficient to fully evaluate the performance of large language models (LLMs) across all real-world scenarios, R1 and R2 noted that the benchmark is more reflective of real-world conditions compared to previous benchmarks. R4 also considered the dataset to be highly comprehensive, offering fine-grained evaluation. The majority consensus leans towards accepting the paper. The authors should carefully consider the reviewers' comments and suggestions to enhance the paper's quality and address the concerns raised. Implementing these revisions in the final version will further improve the manuscript.